# Nearly-Linear Time Private Hypothesis Selection with the Optimal Approximation Factor

**Maryam Aliakbarpour**[1,2], **Zhan Shi**[1], **Ria Stevens**[1], **Vincent X. Wang**[1]
[1]Rice University, Department of Computer Science [2]Ken Kennedy Institute
{maryama, zs50, ria.stevens, vw12}@rice.edu

## Abstract

Estimating the density of a distribution from its samples is a fundamental problem in statistics. *Hypothesis selection* addresses the setting where, in addition to a sample set, we are given $n$ candidate distributions—referred to as *hypotheses*—and the goal is to determine which one best describes the underlying data distribution. This problem is known to be solvable very efficiently, requiring roughly $O(\log n)$ samples and running in $\tilde{O}(n)$ time. The quality of the output is measured via the total variation distance to the unknown distribution, and the approximation factor of the algorithm determines how large this distance is compared to the optimal distance achieved by the best candidate hypothesis. It is known that $\alpha = 3$ is the optimal approximation factor for this problem. We study hypothesis selection under the constraint of *differential privacy*. We propose a differentially private algorithm in the central model that runs in nearly-linear time with respect to the number of hypotheses, achieves the optimal approximation factor, and incurs only a modest increase in sample complexity, which remains polylogarithmic in $n$. This resolves an open question posed by [Bun, Kamath, Steinke, Wu, NeurIPS 2019]. Prior to our work, existing upper bounds required quadratic time.

## 1 Introduction

The task of accurately estimating the underlying probability distribution that generates a dataset is a fundamental theoretical problem in statistical inference with broad applicability in practical data analysis. A growing concern in modern data analysis is preserving the privacy of the individuals whose data informs these estimations, specifically when dealing with sensitive information. *Differential privacy* (DP) has emerged as a widely adopted standard in privacy-preserving data analysis [DMNS06a] and is currently employed by major entities, such as Google [EPK14], Apple [Dif17], and the U.S. Census Bureau [Abo18]. See Section A.1 for more examples.

In this paper, we study a specific instance of distribution estimation under the constraint of differential privacy, referred to as *Hypothesis Selection*. In this problem, we are given a finite collection of $n$ *candidate distributions* $\mathcal{H} \coloneqq \{H_1, H_2, \ldots, H_n\}$, known as hypotheses, and a dataset of i.i.d. samples drawn from an *unknown* distribution $P$. The goal is to select a hypothesis in $\mathcal{H}$ that well-approximates the true data distribution.

A long line of research has studied hypothesis selection in the non-private setting [Yat85, DL96, DL97, DL01, MS08, DK14, SOAJ14, AJOS14, AFJ+18, BBK+21, ABS23, AAC+23, AAC+24, ABS24]. These algorithms' performances are evaluated across three key aspects: *i) sample complexity, ii) time complexity, and iii) approximation factor*. Early work has shown that hypothesis selection admits highly sample-efficient algorithms, requiring only $\Theta(\log n)$ samples, a logarithmic dependence on the number of hypotheses, and no dependence on the domain size of distributions [Yat85, DL96, DL97, DL01]. The sample efficiency is achieved by only needing to estimate probabilities of $O(n^2)$ special sets, called Scheffé sets, according to $P$ (see Equation (1) for a definition). Moreover, several

works [DK14, AFJ+18, ABS23, ABS24] have shown that one can find a valid hypothesis in roughly linear time in $n$. Recently, [AAC+24] characterized the statistical-computational trade-off of the problem when the distributions have finite domain.

Another key aspect is the accuracy of the selected hypothesis, measured by the total variation distance to the true distribution. The approximation factor (denoted by $\alpha$) measures this distance relative to the minimum distance between $P$ and a hypothesis in $\mathcal{H}$. A notable lower bound established in [BKM19] shows that achieving $\alpha < 3$ is impossible unless the number of samples is polynomial in the domain size of $P$. In an interesting development, Aliakbarpour et al. [ABS24] recently proposed an algorithm that simultaneously achieves a logarithmic sample complexity, nearly-linear time complexity, and the optimal approximation factor $\alpha = 3$, representing a compelling performance in all three critical aspects of this problem.

Despite this desirable performance in the non-private setting, the state of the art in the private setting falls short of optimal performance. A naive privatization of Scheffé estimates via Laplace noise leads to an $O(n^2)$ sample complexity [BKSW19]. More advanced techniques, including the work of Bun et al. [BKSW19], offer better sample complexity (logarithmic) but suffer from quadratic time and suboptimal approximation. While Aden-Ali et al. [AAAK21] makes progress on accuracy and proof simplicity, their algorithm remains computationally expensive with a quadratic time complexity.

These limitations naturally lead us to address an open question, first raised in part by Bun et al. [BKSW19]: *Does there exist an ideal private hypothesis selection algorithm that offers logarithmic sample complexity, nearly-linear time complexity, and the optimal approximation factor?* We present a significant step forward towards this ideal: our algorithm achieves nearly-linear time complexity and the optimal approximation factor with a polylogarithmic sample complexity (which, while not logarithmic, is still considered a modest dependence on the number of hypotheses). A formal description of our results can be found in Section 1.2.

**Applications:** Estimating data distributions is a central component of many scientific tasks, such as estimating species abundance in ecology or analyzing survey results in the social sciences. Hypothesis selection describes a broadly applicable scenario where we can form a finite set of interpretable, noise-free, or otherwise manageable distributions as our candidate hypotheses, and we aim to approximate the potentially noisy and complex unknown distribution with one of the so-called "nicer" candidates (e.g., modeling customer arrival time with Poisson processes).

One notable theoretical application of hypothesis selection is agnostic learning of a parametric class of distributions (e.g., mixtures of Gaussians [SOAJ14, DK14, ABM18, ABH+20], and junta distributions [ABR16]) via the *cover method*. The approach is to first select a representative set of parametric distributions, and hypothesis selection then identifies the closest approximation in this set. For a survey, see Diakonikolas [Dia16].

## 1.1 Problem setup

Let $P$ denote an unknown distribution over a domain $\mathcal{X}$, and let $\mathcal{H} := \{H_1, H_2, \ldots, H_n\}$ be a set of $n$ public and known distributions over $\mathcal{X}$. We define OPT to be the total variation distance of $P$ and the closest hypothesis in $\mathcal{H}$.

We seek to design a semi-agnostic proper learner such that for every $\mathcal{H}$ and $P$, the algorithm outputs a hypothesis $\hat{H} \in \mathcal{H}$ such that the total variation distance between $\hat{H}$ and $P$ is within $\alpha$-times OPT plus an additive error parameter $\sigma$, which can be made arbitrarily small with sufficiently many samples.

We assume the standard access model of [DL01], where an algorithm accesses distributions by making queries of the following types:

1. The algorithm can draw i.i.d. samples from the unknown distribution $P$.

2. The algorithm can compare the PDFs of any two known distributions $H_i, H_j$ at a given point $x \in \mathcal{X}$. Specifically, it can ask if $H_i(x) < H_j(x)$. This is equivalent to determining if $x$ is in the Scheffé set of $H_i$ and $H_j$ (defined in Equation 1).

3. The algorithm can query the probability mass of the Scheffé set of any two known distributions. Precisely, it can ask for $H_i(\mathcal{S}_{i,j})$ for all $H_i, H_j \in \mathcal{H}$.

More formally, we have:

**Definition 1.1** (Proper learner for private hypothesis selection). *Let $\alpha > 0$, and let $\mathcal{A}$ be an algorithm with input parameters $\epsilon, \beta, \sigma \in (0, 1)$, sample access to an unknown distribution $P$, and query access to a finite class of $n$ hypotheses $\mathcal{H} = \{H_1, H_2, \ldots, H_n\}$ (according to the access model described above). We say $\mathcal{A}$ is an $(\alpha, \epsilon, \beta, \sigma)$-proper learner for the private hypothesis selection problem if:*

1. *$\mathcal{A}$ is $\epsilon$-differentially private in the* central model *(defined in Definition 1.2) with respect to the samples drawn from $P$.*

2. *$\mathcal{A}$ outputs $\hat{H} \in \mathcal{H}$ such that with probability at least $1 - \beta$,*

$$\|\hat{H} - P\|_{TV} \leq \alpha \cdot \mathrm{OPT} + \sigma.$$

*We call $\alpha$ the approximation factor, $\epsilon$ the privacy parameter, $\beta$ the confidence parameter, and $\sigma$ the error parameter.*

**Remark 1.** *As mentioned in [DK14, ABS24], the third type of query in the standard access model can be relaxed. Our algorithms only need* estimates, *rather than exact values, of the probability masses of the Scheffé sets. Assuming all estimates of $H_i(\mathcal{S}_{i,j})$ are accurate to an additive error of $O(\sigma)$ with high probability, the analysis of our algorithms remain essentially unchanged. These estimates could be obtained by sampling from each $H_i$ or numerically integrating density functions when analytic forms are available.*

## 1.2 Our result

We present an algorithm for private hypothesis selection with the following guarantee:

**Theorem 2** (Informal version of Theorem 3). *For every $\epsilon, \beta, \sigma \in (0, 1)$, Algorithm 1 is an $(\alpha = 3, \epsilon, \beta, \sigma)$-proper learner for the private hypothesis selection problem that uses $s = \Theta(\log^3(n/\beta) / (\beta^2 \sigma^2 \epsilon))$ samples and runs in time $\tilde{\Theta}(n / (\beta^4 \sigma^3 \epsilon))$.*

Our result is the first algorithm for private hypothesis selection that runs in nearly-linear time, resolving an open question first posed by [BKSW19] about the existence of such an algorithm. In addition, we also maintain the optimal approximation factor $\alpha = 3$. However, our algorithm introduces an additional factor of $O(\log^2 n / \sigma)$ in the sample complexity compared to existing work on private hypothesis selection. We summarize this tradeoff and compare with existing private algorithms in Table 1.

The overhead in the sample complexity stems from our privatization strategy that is informed by the structure of our algorithm. Similar to most time efficient algorithms in previous works, our algorithm consists of multiple interdependent components, where each component directs us to focus only on a small set of Scheffé estimates, as opposed to computing all of them. While these interdependencies allow us to achieve a highly time-efficient algorithm, they make direct privatization of the final output analytically very challenging. Rather than attempting to analyze the privacy loss of the full computation, we enforce differential privacy at each component of the algorithm, resulting in the sub-optimality of our sample complexity. Despite this added overhead, our algorithm still maintains sample complexity polylogarithmic in number of hypotheses with no dependence on the domain size.

Our algorithm also has a polynomial dependence on $1/\beta$, contrasting with the typical $\log(1/\beta)$ dependence on the confidence parameter that arises in many learning theory problems when amplifying a result (e.g., by taking an "average" of the results of running $\log(1/\beta)$ many runs of an algorithm). However, in hypothesis selection, this amplification process is unlikely to succeed without a significant increase in $\alpha$ because it would require running hypothesis selection *twice*, which would push the total approximation factor to at least $\alpha = 9$. Achieving a polylogarithmic dependence on $\beta$ is a difficult task, even in the absence of privacy constraints; both [ABS23, ABS24] present nearly-linear time algorithms with $\alpha < 9$ but suffer from similar polynomial dependencies on $1/\beta$.

**Open directions:** This leaves two open directions for further work: *i)* Does there exist a nearly-linear time algorithm that uses $O\left(\frac{\log n}{\sigma^2} + \frac{\log n}{\sigma \epsilon}\right)$ samples in terms of its dependence on $\sigma, \epsilon$? *ii)* Can the dependence on the confidence parameter $\beta$ be improved to $O(\log(1/\beta))$ while maintaining nearly-linear runtime?

Table 1: Summary of past hypothesis selection results under central DP

| Result | $\alpha$ | Sample complexity ($s$) | Time complexity |
|---|---|---|---|
| Private Scheffé tournament [BKSW19] (Thm 3.6) | 9 | $O\left(\frac{\log n}{\sigma^2} + \frac{n^2 \log n}{\sigma\epsilon}\right)$ | $O(n^2 \cdot s)$ |
| [BKSW19] (Thm 3.5) | $> 54$ | $\tilde{O}\left(\frac{\log n}{\sigma^2} + \frac{\log n}{\sigma\epsilon}\right)$ | $\tilde{O}(n^2 \cdot s)$ |
| Minimum distance estimate [AAAK21] (Thm 2.24) | 3 | $O\left(\frac{\log n}{\sigma^2} + \frac{\log n}{\sigma\epsilon}\right)$ | $O(n^2 \cdot s)$ |
| This work | 3 | $O\left(\frac{\log^3 n}{\sigma^2 \epsilon}\right)$ | $\tilde{O}\left(n \cdot s / \sigma\right)$ |

**Significance of the approximation factor:** One may argue that it is possible to improve the accuracy by decreasing OPT by selecting more candidate hypotheses in $\mathcal{H}$, as opposed to decreasing the approximation factor $\alpha$. However, as mentioned in [ABS24], this is not feasible in many spaces, especially multivariate distributions, without substantially increasing the size of $\mathcal{H}$. For instance, in the cover method, we require $\mathcal{H}$ to cover the space of a parametric class of distributions with a $\gamma$-cover, where each distribution in the class is at most $\gamma$ away from an element in $\mathcal{H}$, enforcing OPT $< \gamma$. As an example, the mixture of $k$-Gaussians in [SOAJ14] requires $O(\gamma^{-3k})$ distributions to create a $\gamma$-cover, making algorithms with high approximation factor more time-consuming.

## 1.3 Related work

Most of the previous work on hypothesis selection falls under two approaches: ***i)*** *tournament-based* algorithms that compare pairs of hypotheses based on their Scheffé estimates, and then perform a series of comparisons to find a final winner ***ii)*** the *minimum distance estimate (MDE)* algorithms that compute an *approximate distance* based on the Scheffé estimates for each hypothesis, and then select the hypothesis with the minimum approximate distance.

The Scheffé tournament algorithm, first proposed by Devroye and Lugosi [DL01], runs in $O(n^2 \cdot s)$ time and achieves an approximation factor of $\alpha = 9$. Other works in this line include [DK14, AJOS14, AFJ+18, AAC+23, ABS23]. Algorithms that follow the tournament structure typically exhibit a high approximation factor. Moreover, privatizing such algorithms raises extra challenges because changing a single data entry might affect the result of every comparison, thus greatly increasing sensitivity.

The other approach [DL01, MS08, ABS24] for non-private hypothesis selection is based on the *minimum distance estimate* (MDE) method introduced in [DL01], which has been the only type of algorithm that achieves the optimal approximation factor $\alpha = 3$. Mahalanabis and Stefankovic [MS08] later improved the initial $O(n^3 \cdot s)$ runtime of [DL01] to $O(n^2 \cdot s)$, as well as proposed a nearly-linear time algorithm that requires exponential pre-processing. Aliakbarpour et al. [ABS24] recently demonstrated that the optimal approximation factor $\alpha = 3$ could be achieved with a nearly-linear time algorithm.

For an extended version of the related work, see Appendix A.

## 1.4 Background: differential privacy

**Differential privacy:** We adopt the *central* model of DP [DMNS06b], where a sensitive dataset is given to a trusted data curator who performs the algorithm and publicly publishes the outcome. Differential privacy protects each entry in the dataset from an adversary who observes the outcome.

A dataset $D = [x_1, \ldots, x_s] \in \mathcal{X}^{\otimes s}$ is a collection of $s$ i.i.d. samples from an unknown distribution $P$. The *Hamming distance* between two datasets $D$ and $D'$ is defined as the number of differing entries and denoted as $\mathbf{Ham}(D, D')$. We consider an algorithm to be private with respect to the samples drawn from $P$ if it satisfies the following definition:

**Definition 1.2** (Pure differential privacy). *Let $\epsilon > 0$. An algorithm $\mathcal{A}$ is $\epsilon$-differentially private if for all measurable subsets $\mathcal{S} \subseteq \mathrm{Range}(\mathcal{A})$ and $D, D' \in \mathcal{X}^{\otimes s}$ such that $\mathbf{Ham}(D, D') = 1$:*

$$\mathbf{Pr}[\mathcal{A}(D) \in \mathcal{S}] \leq e^\epsilon \mathbf{Pr}[\mathcal{A}(D') \in \mathcal{S}].$$

Standard methods for calibrating noise in DP rely on the concept of sensitivity:

**Definition 1.3** (Sensitivity). *Let $f : \mathcal{X}^{\otimes s} \to \mathbb{R}$ be a function. Then, $\Delta(f)$ denotes the* sensitivity *of $f$ and is defined by:*

$$\Delta(f) := \sup_{\substack{D,D' \in \mathcal{X}^{\otimes s} \\ \mathbf{Ham}(D,D')=1}} |f(D) - f(D')|.$$

**Exponential mechanism:** A widely used algorithm in DP is the exponential mechanism. This mechanism relies on a real-valued utility function $u$ that maps a dataset $D \in \mathcal{X}^{\otimes s}$ and a candidate output $H_j \in \mathcal{H}$ to a real-valued score, quantifying the "quality" of $H_j$ with respect to $D$. Outputs with higher utility are more likely to be selected.

**Definition 1.4** (Exponential mechanism [MT07, DR$^+$14]). *Given a utility function $u : \mathcal{X}^{\otimes s} \times \mathcal{H} \to \mathbb{R}$ with sensitivity $\Delta(u)$, the exponential mechanism selects an element $H_j \in \mathcal{H}$ with probability proportional to $\exp\left(\frac{\epsilon u(D,H_j)}{2\Delta(u)}\right)$.*

**Fact 1.5** ([MT07]). *The exponential mechanism is $\epsilon$-differentially private.*

We also have the following utility guarantee of the exponential mechanism:

**Fact 1.6** (Corollary 3.12 of [DR$^+$14]). *Let $u : \mathcal{X}^{\otimes s} \times \mathcal{H} \to \mathbb{R}$ be a utility function with sensitivity $\Delta(u)$. Fix a dataset $D \in \mathcal{X}^{\otimes s}$. Let $H \in \mathcal{H}$ denote the output of the exponential mechanism with parameter $\epsilon$ and utility function $u$. Then, for any $\beta \in (0,1)$:*

$$\mathbf{Pr}\left[u(D,H) \leq \max_{H_j \in \mathcal{H}} u(D,H_j) - \frac{2\Delta(u)}{\epsilon} \log(n/\beta)\right] \leq e^{-\log(1/\beta)} = \beta.$$

**Sparse vector technique [DNR$^+$09, DR$^+$14, LSL16]:** Given a numerical statistic $g : \mathcal{X} \to \mathbb{R}$, a *threshold query* counts the number of entries $x \in D$ such that $g(x)$ is above or below a fixed cutoff. We will employ the sparse vector technique (SVT) [DNR$^+$09], which enables processing a large number of threshold queries while incurring a privacy cost only for the small subset of queries whose values exceed a specified threshold. Dwork et al. [DNR$^+$09] presents a simple $\epsilon$-differentially private algorithm ABOVETHRESHOLD that takes in a stream of queries and identifies the first meaningful query above a predefined threshold while privately ignoring queries that fall below the threshold.

**Composition and post-processing:** Two properties make DP particularly well-suited for modular algorithm design. *Composition* bounds the cumulative privacy loss after performing multiple differentially private subroutines, and *post-processing* ensures that no further transformation of the output of a differentially private algorithm can further degrade the privacy guarantees. We state the following theorems:

**Fact 1.7** (Composition, Theorem 3.14 of [DR$^+$14]). *Let $\mathcal{A}_1, \ldots, \mathcal{A}_k$ be algorithms that access the same dataset $D \in \mathcal{X}^{\otimes s}$, and suppose each $\mathcal{A}_i$ is $\epsilon_i$-differentially private. Let $\mathcal{A}$ be the composed algorithm of $\mathcal{A}_1, \ldots, \mathcal{A}_k$. Then, $\mathcal{A}$ is $\sum_{i=1}^{k} \epsilon_i$-differentially private.*

**Fact 1.8** (Post-processing, Proposition 2.1 of [DR$^+$14]). *Let $\mathcal{A}$ be an $\epsilon$-differentially private algorithm. Let $g$ be a (possibly random) mapping. Then, $g \circ \mathcal{A}$ is $\epsilon$-differentially private.*

## 2 Preliminaries

We begin by introducing notation and a list of key definitions in Section 2.1. These concepts are expanded upon in later sections. Section 2.2 presents the framework of minimum distance estimate algorithms, and Section 2.3 describes the nearly-linear time optimization proposed by [ABS24].

### 2.1 Notation and basic concepts

**Notation and basic definitions:** For $n \in \mathbb{Z}_+$, we use $[n]$ to denote the set $\{1, \ldots, n\}$. For an arbitrary probability distribution $P$ over $\mathcal{X}$, let $P(x)$ be the PDF of $P$ at $x \in \mathcal{X}$. For a measurable subset $S \subseteq \mathcal{X}$, let $P(S)$ be the probability mass of the set $S$ according to $P$. We use $X \sim P$ to denote a random variable $X$ that is drawn from the distribution $P$. Let $\|P_1 - P_2\|_{\mathrm{TV}} := \sup_{S \subseteq \mathcal{X}} |P_1(S) - P_2(S)|$ be the total variation distance between two distributions $P_1$ and $P_2$. For a sample space $\Omega$

and an event $E \subseteq \Omega$, the indicator function $\mathbb{1}_E$ evaluates to 1 when $E$ occurs and 0 otherwise. We use the standard $O, \Omega, \Theta$ notation for asymptotic functions, as well as $\tilde{O}(x), \tilde{\Omega}(x), \tilde{\Theta}(x)$ to indicate additional polylog$(x)$ factors. A dataset $D = [x_1, \ldots, x_s] \in \mathcal{X}^{\otimes s}$ is a collection of $s$ i.i.d. samples from an unknown distribution $P$.

**Optimal hypothesis:** We use $H_{i^*}$ to indicate a hypothesis in a finite hypothesis class $\mathcal{H}$ that achieves the smallest total variation distance to $P$, which is called OPT. If there are ties, then we pick one such hypothesis as $H_{i^*}$ arbitrarily. Therefore, OPT $:= \min_{H \in \mathcal{H}} \|H - P\|_{\mathrm{TV}} = \|H_{i^*} - P\|_{\mathrm{TV}}$.

**Scheffé sets:** For every pair of hypotheses $H_i, H_j \in \mathcal{H}$, we define the *Scheffé set* of $H_i$ and $H_j$ as:

$$\mathcal{S}_{i,j} := \begin{cases} \{x \in \mathcal{X} \mid H_i(x) < H_j(x)\} & \text{if } i \leq j, \\ \mathcal{S}_{j,i} & \text{if } i > j. \end{cases} \tag{1}$$

It is not difficult to show that the difference of probability masses of two distributions on the Scheffé set of $H_i$ and $H_j$ is precisely the total variation distance between $H_i$ and $H_j$:

$$\|H_i - H_j\|_{\mathrm{TV}} = \sup_{S \subseteq \mathcal{X}} |H_i(S) - H_j(S)| = |H_i(\mathcal{S}_{i,j}) - H_j(\mathcal{S}_{i,j})|.$$

**Semi-distances:** We adopt the definitions of semi-distances from [ABS24], building on earlier work in [DL01, MS08]. For every pair of hypotheses $H_i, H_j \in \mathcal{H}$, the *semi-distance* $w_i(H_j)$ is the distance between $H_j$ and $P$ measured on the Scheffé set of $H_i$ and $H_j$; that is, $w_i(H_j) := |H_j(\mathcal{S}_{i,j}) - P(\mathcal{S}_{i,j})|$. The *maximum semi-distance of $H_j$* is defined as $W(H_j) := \max_{H_i \in \mathcal{H}} w_i(H_j)$.

**Empirical semi-distances:** Given a measurable set $S \subseteq \mathcal{X}$, we define the empirical distribution $\hat{P}$ of a dataset $D = [x_1, \ldots, x_s]$ as $\hat{P}(S) := \frac{1}{s} \sum_{k=1}^{s} \mathbb{1}_{x_k \in S}$. The *empirical semi-distance* $\hat{w}_i(H_j)$ is similarly defined as $\hat{w}_i(H_j) := |H_j(\mathcal{S}_{i,j}) - \hat{P}(\mathcal{S}_{i,j})|$, where $\hat{P}$ is based on the observed samples drawn from $P$. Observe that $\hat{w}_i(H_j)$ is an estimation of $w_i(H_j)$. For a given hypothesis $H_j \in \mathcal{H}$ and a set of hypotheses $A \subseteq \mathcal{H}$, the *proxy distance* $\hat{W}_A(H_j)$ is defined as $\hat{W}_A(H_j) := \max_{H_i \in A} \hat{w}_i(H_j)$. Here, $A$ is a set of hypotheses that is updated throughout the algorithm to improve $\hat{W}_A(H_j)$ as an approximation for $W(H_j)$.

**Refined access model:** As in prior work [DL01, MS08, ABS24], our algorithms will have query access to $\hat{w}_i(H_j)$, which follows from the standard access model. In the lemma below, we show that $\hat{w}_i(H_j)$ is within some $\sigma'$ of $w_i(H_j)$ with sufficiently large samples, where $\sigma'$ can be taken to be $\Theta(\sigma)$. The time complexity of our algorithms is measured in number of queries to $\hat{w}_i(H_j)$, and each query takes $\Theta(s)$ to compute.

**Lemma 2.1.** *Let $\beta, \sigma' \in (0, 1)$. If the number of samples $s \geq \frac{1}{2\sigma'^2} \log(2n/\beta)$, then with probability at least $1 - \beta$, the empirical semi-distances are accurate to an additive error of $\sigma'$:*

$$|\hat{w}_i(H_j) - w_i(H_j)| \leq \sigma', \qquad \text{for all } i, j \in [n].$$

*Proof.* The estimates $\hat{w}_i(H_j) = |H_j(\mathcal{S}_{i,j}) - \hat{P}(\mathcal{S}_{i,j})|$ can be computed via sampling from $P$ and counting the fraction of samples in the Scheffé set of $H_i$ and $H_j$, which is $\hat{P}(\mathcal{S}_{i,j})$. By a reverse triangle inequality:

$$|\hat{w}_i(H_j) - w_i(H_j)| = \left| |H_j(\mathcal{S}_{i,j}) - \hat{P}(\mathcal{S}_{i,j})| - |H_j(\mathcal{S}_{i,j}) - P(\mathcal{S}_{i,j})| \right| \leq |\hat{P}(\mathcal{S}_{i,j}) - P(\mathcal{S}_{i,j})|.$$

Therefore, by a standard application of the Hoeffding and union bounds, we can estimate each $w_i(H_j)$ using $\frac{1}{2\sigma'^2} \log(2n/\beta)$ samples. $\qquad\square$

**Lifting:** Let $H_i, H_j \in \mathcal{H}$. We define the *lift value $H_i$ induces on $H_j$* by $\hat{w}_i(H_j) - \hat{W}_A(H_j)$. In other words, the lift value quantifies the improvement of the proxy distance $\hat{W}_A(H_j)$ if $H_i$ is added to the set of hypotheses $A$ used to compute $\hat{W}_A(H_j) = \max_{H_k \in A} w_k(H_i)$. For some $\sigma' \in (0, 1)$, we say that $H_i$ $\sigma'$-*lifts* $H_j$ if the lift value is at least $\sigma'$, or equivalently that $H_i$ lifts $H_j$ by at least $\sigma'$.

**Prompting:** Let $Q$ be a distribution over $\mathcal{H}$. For two parameters $\sigma', \eta \in (0,1)$, we say a hypothesis $H_i \in \mathcal{H}$ is $(\sigma', \eta)$-*prompting with respect to* $Q$ if $H_i$ $\sigma'$-lifts a random hypothesis $H_j$ sampled from $Q$ with probability at least $\eta$. In other words, we have:

$$\mathbf{Pr}_{H_j \sim Q}\left[\hat{w}_i(H_j) - \hat{W}_A(H_j) \geq \sigma'\right] \geq \eta. \tag{2}$$

We now consider an empirical analog for a list of hypotheses $\mathcal{K}$ that is sampled from $Q$. Let $\mathcal{K} = [H_{j_1}, \ldots, H_{j_t}]$ be a list of $t$ hypotheses in $\mathcal{H}$. For two parameters $\sigma', \eta \in (0,1]$, we say a hypothesis $H_i \in \mathcal{H}$ is $(\sigma', \eta)$-*empirical-prompting with respect to* $\mathcal{K}$ if $H_i$ $\sigma'$-lifts at least an $\eta$-fraction of the hypotheses in $\mathcal{K}$. In other words, we have:

$$\frac{1}{t} \sum_{k=1}^{t} \mathbb{1}_{\{\hat{w}_i(H_{j_k}) - \hat{W}_A(H_{j_k}) \geq \sigma'\}} \geq \eta.$$

## 2.2 Background: minimum distance estimate

In this section, we sketch the key ideas behind previous approaches that use a *minimum distance estimate* [DL01, MS08]. For a more detailed treatment, see Section 3.1 of [ABS24].

Our algorithms rely on computations of $\hat{w}_i(H_j)$ that approximate the true semi-distances $w_i(H_j)$. For clarity, we will assume in this section that the approximations $\hat{w}_i(H_j)$ are exact by Lemma 2.1. Observe that $w_i(H_j)$ provides a lower bound for $\|H_j - P\|_{\text{TV}}$: specifically, $w_i(H_j) = |H_j(\mathcal{S}_{i,j}) - P(\mathcal{S}_{i,j})| \leq \sup_{S \subseteq \mathcal{X}} |H_j(S) - P(S)| = \|H_j - P\|_{\text{TV}}$. Thus, we can view $w_i(H_j)$ as an attempt to lower bound $\|H_j - P\|_{\text{TV}}$. In particular, if we discover that $w_i(H_j)$ is large, then this suggests that $H_j$ is far from $P$. However, the inverse is not true: when $w_i(H_j)$ is small, this does not imply that $H_j$ is close to $P$.

Fortunately, when the particular semi-distance $w_{i^*}(H_j)$ is small, we can upper bound $\|H_j - P\|_{\text{TV}}$. The semi-distance $w_{i^*}(H_j)$ has the following property: if $w_{i^*}(H_j) \leq \text{OPT}$, then $H_j$ satisfies $\|H_j - P\|_{\text{TV}} \leq 3 \cdot \text{OPT}$, making $H_j$ a valid hypothesis to output. This follows from repeated applications of the triangle inequality:

$$\begin{aligned}
\|H_j - P\|_{\text{TV}} &\leq \|H_j - H_{i^*}\|_{\text{TV}} + \|H_{i^*} - P\|_{\text{TV}} = |H_{i^*}(\mathcal{S}_{i^*,j}) - H_j(\mathcal{S}_{i^*,j})| + \text{OPT} \\
&= |(H_{i^*}(\mathcal{S}_{i^*,j}) - \hat{P}(\mathcal{S}_{i^*,j})) - (H_j(\mathcal{S}_{i^*,j}) - \hat{P}(\mathcal{S}_{i^*,j})| + \text{OPT} \\
&\leq w_{i^*}(H_j) + w_j(H_{i^*}) + \text{OPT} \leq w_{i^*}(H_j) + 2 \cdot \text{OPT}. \tag{3}
\end{aligned}$$

An issue with using this metric ($w_{i^*}(H_j) \leq \text{OPT}$) is that we know neither the optimal hypothesis $H_{i^*}$ nor OPT. This is remedied by minimizing a *maximum semi-distance* $W(H_j) := \max_{H_i \in \mathcal{H}} w_i(H_j)$. Let $\hat{H}$ be the hypothesis that minimizes $W(H_j)$ over all hypotheses in $\mathcal{H}$. Then, we claim that $\hat{H}$ satisfies the desired property $w_{i^*}(\hat{H}) \leq \text{OPT}$, implying that $\|\hat{H} - P\|_{TV} \leq 3 \cdot \text{OPT}$ as in the above discussion. This follows from:

$$w_{i^*}(\hat{H}) \leq W(\hat{H}) \leq W(H_{i^*}) \leq \text{OPT}.$$

The first inequality follows from $W(\hat{H})$ being a maximum over all semi-distances of $\hat{H}$. The second inequality follows from minimality of $W(\hat{H})$ over all other hypotheses. The last inequality follows from the fact that every semi-distance measured against the optimal hypothesis is itself bounded by OPT. Therefore, we reduce the problem of hypothesis selection to finding a $\hat{H}$ that minimizes $W(\hat{H})$.

## 2.3 Background: approximating the maximum semi-distance

The MDE framework is sample-optimal and achieves an optimal error parameter of $\alpha = 3$. However, computing all $W(H_j)$'s exactly requires $\tilde{O}(n^2)$ time, which is expensive for large hypotheses classes. Instead, [ABS24] computes a *proxy distance* $\hat{W}_A(H_j) = \max_{H_k \in A} \hat{w}_k(H_j)$ for each $H_j$, which serves as an updateable approximation that lower bounds $W(H_j)$. All $\hat{W}_A(H_j)$'s are initially set to 0. The proxy distances are updated throughout the algorithm via an iterative process. At every iteration, a selectively chosen hypothesis, called a *prompting hypothesis*, is added to $A$. Carefully selecting $A$

ensures that for all $H_j$, $\hat{W}_A(H_j)$ is a good approximation of $W(H_j)$ without exhaustively computing all pairwise semi-distances. At the end of this process, a hypothesis with a low proxy distance is selected as the output. Because only $O(|A| \cdot n)$ semi-distances are queried, this strategy enables a nearly-linear time algorithm in $\tilde{O}(n)$ that still maintains the $\alpha = 3$ guarantee.

To identify prompting hypotheses, [ABS24] keeps track of "buckets" of candidate hypotheses. Each hypothesis $H_j$ is assigned a bucket according to its proxy distance $\hat{W}_A(H_j)$. Because the algorithm seeks a hypothesis with approximately the smallest maximum semi-distance, it only focuses on the "lowest" bucket with the smallest proxy distances. Because a large proxy distance implies that the hypothesis is far from $P$, the algorithm can disregard hypotheses with a large proxy distance. Hence, only the hypothesis in the lowest bucket are required to have accurate proxy distances.

The set of prompting hypotheses $A$ will ensure that the lowest bucket has hypotheses with proxy distances close to the maximum semi-distances.

Recall that if a hypothesis $\hat{H}$ satisfies $w_{i^*}(\hat{H}) \leq \text{OPT}$, then it is a valid output. Conversely, we would like to avoid outputting a hypothesis for which we have $w_{i^*}(\hat{H}) > \text{OPT}$. The algorithm of [ABS24] filters out such hypotheses in the lowest bucket by updating their proxy distance and effectively sending them to "higher" buckets. Specifically, a key observation is that $H_{i^*}$ will always significantly lift the proxy distances of hypotheses that are poor choices. Therefore, in every iteration of the algorithm, the goal is to find a *prompting* hypothesis $H_i$ that substantially improves a large portion of the proxy distances and empties out the candidate hypotheses in the lowest bucket.

It can be shown after roughly $\Theta(\log n)$ iterations either the lowest bucket is fully emptied out, and the algorithm can move forward to the next bucket. Or, there are no more prompting hypotheses that can be identified. This condition implies that $H_{i^*}$ is not prompting. That is, $H_{i^*}$ could not lift most hypotheses in the lowest bucket. For those hypotheses, we must have that their $w_{i^*}(H_j) \leq \text{OPT}$. Hence, a random hypothesis in the lowest bucket under this condition is a valid output.

## 3 Private Hypothesis Selection Algorithm

### 3.1 Overview of our algorithm

In this section, we present an overview of Algorithm 1, our main algorithm for solving the hypothesis selection problem in the central model of DP that obtains an $\alpha = 3$ guarantee. Building on the previous work described in Section 2.2, our goal is to find a hypothesis $\hat{H}$ that approximately minimizes $W(\hat{H})$. For each hypothesis $H_j \in \mathcal{H}$, our algorithm keeps track of the proxy distances $\hat{W}_A(H_j) = \max_{H_k \in A} \hat{w}_k(H_j)$. The set $A$ will store a small set of prompting hypotheses accumulated so far. In every round $t$ of our algorithm, we privately identify a prompting hypothesis $H_{i_t}$ that improves a substantial portion of current proxy distances $\hat{W}_A(H_j)$ and update each $\hat{W}_A(H_j)$ to $\max\left(\hat{W}_A(H_j), \hat{w}_{i_t}(H_j)\right)$. This effectively adds the privately selected $H_{i_t}$ to the set $A$.

The primary challenge of privatizing the process of identifying a prompting hypothesis in [ABS24] arises due to the bucketing scheme. This membership to a bucket is highly sensitive due to its discrete nature. In particular, changing one sample of the dataset could potentially shift the membership of every single hypothesis in $\mathcal{H}$ by changing every $\hat{W}_A(H_j)$.

Therefore, to privately select hypotheses with low proxy distances without relying on buckets, we use the exponential mechanism. We maintain a distribution of hypotheses $Q$ that assigns each hypothesis $H_j$ a probability that favors hypotheses with low proxy distance $\hat{W}_A(H_j)$. Based on this change, we modify the notion of prompting from [ABS24]: we say that a hypothesis is prompting with respect to the *distribution* $Q$ over $\mathcal{H}$, rather than with respect to the hypotheses in a bucket. We also introduce the notion of $(\sigma', \eta)$-prompting to quantify the "prompting-ness" of a hypothesis in Equation 2, where $\eta$ is the probability mass of hypotheses in $Q$ that can be $\sigma'$-lifted. Recall that a hypothesis $H_i$ $\sigma'$-lifts a hypothesis $H_j$ if $\hat{w}_i(H_j) - \hat{W}_A(H_j) \geq \sigma'$. After sufficiently many rounds, when no more prompting hypotheses can be found, we sample an output hypothesis $\hat{H}$ from distribution $Q$.

**Identifying prompting hypotheses:** To privately identify a prompting hypothesis $H_i$, we test whether $H_i$ significantly lifts a large probability mass of hypotheses in $Q$. A straightforward approach

is to first create a list $\mathcal{K}$ of hypotheses that are sampled from $Q$. Then, we may choose a prompting hypothesis using a threshold query for hypotheses in $\mathcal{K}$ that are lifted significantly. Unfortunately, such a threshold query would have a very high sensitivity with respect to the dataset $D$, as a single change in the dataset can shift every lift value from below the threshold to being above the threshold.

To solve the issue of the high sensitivity, we replace the exact count of hypotheses lifted with a new type of query called $\mathsf{score}_{\eta,\mathcal{K},D}(H_i)$. This query instead returns a *quantile* of the lift values of each hypothesis, which is a much more stable type of query with a lower sensitivity.

More specifically, $\mathsf{score}_{\eta,\mathcal{K},D}(H_i)$ is computed by Algorithm 2 as follows: we first calculate the lift value $H_i$ induces on each element in $\mathcal{K}$. Then, we sort these values in non-increasing order and return the $\lceil \eta/2 \cdot |\mathcal{K}| \rceil$-th largest lift value. This significantly reduces the sensitivity of $\mathsf{score}_{\eta,\mathcal{K},D}(H_i)$, as shown in section E.1. Even if every single value in $H_i$ shifts by some amount due to a change in the dataset, the quantile that we return should not shift significantly. In Section E.2, we show that $\mathsf{score}_{\eta,\mathcal{K},D}(H_i)$ can be used to identify whether or not $H_i$ is prompting.

**Applying the SVT:** We now wish to determine exactly which hypotheses have high $\mathsf{score}_{\eta,\mathcal{K},D}(H_i)$'s to find hypotheses that are significantly prompting. For every round of our algorithm, we attempt to find a prompting hypothesis. This task is equivalent to answering $n$ threshold queries. In general, this is very costly from the perspective of privacy. However, because we are only interested in one hypothesis that passes the threshold, we use the sparse vector technique (SVT) [DNR$^+$09] to find this hypothesis with minimal privacy cost.

Algorithm 3 describes an algorithm using the SVT, which privately outputs either the index $i$ of the hypothesis that was detected to have a high $\mathsf{score}_{\eta,\mathcal{K},D}(H_i)$, or $\perp$ if no hypotheses have sufficiently high scores. This algorithm has guarantees on finding a prompting hypothesis formalized in Theorem 6. First, whenever the SVT returns a hypothesis, we guarantee that it is at least somewhat prompting, so we make progress at every round by updating the proxy distances. Second, whenever the SVT fails to find any prompting hypotheses, all hypotheses have small $\mathsf{score}_{\eta,\mathcal{K},D}(H_j)$'s and are therefore unlikely to be prompting.

As in [ABS24], $H_{i^*}$ is typically prompting for hypotheses far from $P$. When the SVT is unable to find any prompting hypotheses, it implies that $H_{i^*}$ is not prompting with respect to $Q$. Consequently, a large probability mass of hypotheses in $Q$—namely, those with minimal proxy distances—must have proxy distances that cannot be lifted by $H_{i^*}$. Recall that all the poor hypotheses can be lifted by $H_{i^*}$, so in this case, outputting any random hypothesis in $Q$ will be valid with high probability.

**Number of rounds:** Because we do not allow a hypothesis to be added to the prompting set more than once, our algorithm will certainly halt after at most $n$ rounds. However, this leads to a quadratic bound on the time complexity. We show that the algorithm will halt after $O(\log n)$ rounds, yielding a nearly-linear time complexity.

Arguing about this round complexity is a key hurdle that arises in the private setting. The non-private algorithm in [ABS24] iteratively eliminates hypotheses from buckets. At every round, upon finding a prompting hypothesis, a significant (say a constant) fraction of hypotheses within the bucket have their proxy distances updated, leading to their removal from the bucket. This ensures that even with an initial bucket of all $n$ hypotheses, the algorithm concentrates on this bucket for only $O(\log n)$ iterations. After this, the bucket is either empty or the algorithm halts due to the lack of a prompting hypothesis. However, adapting the notion of prompting to the case where we update only a set of hypotheses with a constant probability mass according to $Q$ fundamentally changes the analysis. Determining the actual fraction of updated hypotheses becomes much more complex. For example, we might update only one hypothesis, since it may hold a constant probability mass under $Q$.

To resolve this issue, we provide a refined analysis of the progress of the exponential mechanism. This analysis relies on the fact that the normalization term in the exponential mechanism's probabilities must decrease with each prompting hypothesis added to the set $A$, as some proxy distances will increase but none can decrease. As a result, hypotheses whose proxy distances do not increase significantly will see their probabilities rise. As these probabilities cannot exceed one, changes to the proxy distances must be able to keep up with the decrease in the normalization term. However, they can only keep up for so long, as each proxy distance is itself upper bounded by one. As a result, we can bound the number of rounds of our algorithm by $O(\log n)$.

**Enforcing privacy:** Throughout every round of the algorithm, we incur two types of privacy costs: one from drawing $|\mathcal{K}|$ hypotheses from the exponential mechanism and another from identifying a prompting hypothesis through the sparse vector technique. As we have discussed above, the types of queries our algorithm makes have low sensitivities with respect to the dataset. In Lemma B.1, we give a complete proof of privacy by using basic additive composition.

## 3.2 Algorithm

In Algorithm 1, we begin by sampling $s$ samples from the unknown distribution to make up a dataset $D$. We initialize $A$, the set of prompting hypotheses to be empty, and the proxy distance of each hypothesis to 0. We then iteratively search, over at most $T$ rounds, for prompting hypotheses to add to $A$. In each round, we re-calculate the probabilities of the exponential mechanism in Line 9 and draw a list $\mathcal{K}$ of $k$ hypotheses from this mechanism in Line 10. We call the FIND-PROMPTING-HYPOTHESIS procedure of Algorithm 3 to identify a hypothesis which is empirically prompting over $\mathcal{K}$ using the sparse vector technique. In Section F, we thoroughly describe this procedure. In Section E.1, we describe the COMPUTE-SCORE procedure used to assign a score to each hypothesis throughout FIND-PROMPTING-HYPOTHESIS. If FIND-PROMPTING-HYPOTHESIS returns a hypothesis, we add that hypothesis to $A$ in Line 13 and update the proxy estimates of all hypotheses in Line 14. If FIND-PROMPTING-HYPOTHESIS returns $\perp$, indicating that it could not find a prompting hypothesis, we break from the "for" loop, draw a hypothesis using the exponential mechanism, and output that final hypothesis.

---

**Algorithm 1** A private algorithm for hypothesis selection

1: **procedure** SELECT-HYPOTHESIS($\mathcal{H}, \epsilon, \sigma, \beta$)

2:     $s \leftarrow \Theta\left(\frac{1}{\beta^2 \sigma^2 \epsilon} \log^3(n/\beta)\right), T \leftarrow \min\left(\Theta\left(\frac{1}{\beta\sigma}\log(n/\beta)\right), n\right), k \leftarrow \Theta\left(\frac{1}{\beta}\log(n/\beta)\right)$

3:     $\epsilon_1 \leftarrow \frac{\epsilon}{2(kT+1)}, \; \epsilon_2 \leftarrow \frac{\epsilon}{2T}$

4:

5:     $D \leftarrow s$ samples drawn from $P$         ▷ we will use these samples to compute semi-distances

6:     $A \leftarrow \emptyset$

7:     $\hat{W}_A(H_j) \leftarrow 0$ for every $H_j \in \mathcal{H}$

8:     **for** $t = 1, \ldots, T$ **do**

9:         $Q(H_j) \propto \exp\left(-\frac{\epsilon_1 \hat{W}_A(H_j)}{2\Delta(\hat{W}_A)}\right)$ for every $H_j \in \mathcal{H}$

10:         $\mathcal{K} \leftarrow k$ hypotheses drawn from $Q$         ▷ sample using exponential mechanism

11:         $H_{i_t} \leftarrow$ FIND-PROMPTING-HYPOTHESIS $\left(\epsilon_2, \frac{2}{s}, \frac{\sigma'}{4}, \frac{\beta}{4}, \mathcal{H} \setminus A, \mathcal{K}, D\right)$    ▷ Algorithm 3

12:         **if** $H_{i_t} \neq \perp$ **then**

13:             $A \leftarrow A \cup \{H_{i_t}\}$                     ▷ add $H_{i_t}$ to prompting set

14:             $\hat{W}_A(H_j) \leftarrow \max\left(\hat{W}_A(H_j), \hat{w}_{i_t}(H_j)\right)$ for every $H_j \in \mathcal{H}$

15:         **else**

16:             **break**                          ▷ failed to find a prompting hypothesis

17:     **return** $\hat{H} \sim Q$ and **halt**

---

**Theorem 3.** *Let $\epsilon, \beta, \sigma \in (0,1)$. Algorithm 1 is an $(\alpha = 3, \epsilon, \beta, \sigma)$-private learner for hypothesis selection that uses $s = \Theta\left(\frac{1}{\beta^2 \sigma^2 \epsilon} \log^3(n/\beta)\right)$ samples and has time complexity $\Theta\left(\min\left(\frac{1}{\beta^4 \sigma^3 \epsilon} \cdot n \cdot \log^5(n/\beta), \; \frac{1}{\beta^3 \sigma^2 \epsilon} \cdot n^2 \cdot \log^4(n/\beta)\right)\right).$*

**Proof sketch:** In Section B, we prove that Algorithm 1 is $\epsilon$-differentially private. In Section C, we prove the correctness of Algorithm 1. Specifically, in Section C.1, we show that each hypothesis added to $A$ will be prompting with high probability. In Section C.2, we show that, if this is the case, then the algorithm will halt after at most $O(\log n)$ rounds. In Section C.3, we show that, if the algorithm halts early, then it will output a valid hypothesis with high probability. In Section C.5, we give exact settings for $s, T$, and $k$ such that our proof of correctness holds. Finally, in Section D, we prove the time complexity of Algorithm 1.

# 4 Acknowledgments

R.S. acknowledges partial support from the Ken Kennedy Institute Research Cluster Fund and the Ken Kennedy Institute Computational Science and Engineering Recruiting Fellowship, funded by the Energy HPC Conference and the Rice University Department of Computer Science.

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

# A  Other Related Works

Hypothesis selection has also been studied in the *local* model of DP [KLN$^+$11], where the data curator is not trusted and thus only has access to the privatized version of users' data. Gopi et al. [GKK$^+$20] showed that the sample complexity in the local model is linear in $n$, exponentially larger than the central DP setting. Specifically, they proved a lower bound of $\Omega\left(\frac{n}{\sigma^2\epsilon^2}\right)$. They also proved two upper bounds of $O\left(\frac{n\log^3 n}{\sigma^4\epsilon^2}\right)$ for non-interactive algorithms and $O\left(\frac{n\log n\log\log n}{\sigma^2\epsilon^2}\right)$ (with $\alpha = 27$) for sequentially interactive algorithms with $O(\log\log n)$ rounds. A recent algorithm by Pour et al. [PAA24] closed the gap between upper and lower bounds by designing a sequentially interactive algorithm using $O\left(\frac{n\log^2 1/\beta}{\sigma^2\min(\epsilon^2,1)}\right)$ samples, with $\alpha = 9$ and $O(\log\log n)$ rounds.

Another related problem in statistics is *simple* hypothesis testing. Given two distributions $P$ and $Q$, and a dataset $D$ sampled from one of them, the goal is to determine whether $D$ was drawn from $P$ or $Q$. This setting resembles hypothesis selection where we have only two candidate distributions. Cannone et al. [CKM$^+$19] developed a private algorithm for simple hypothesis testing that achieves optimal sample complexity. Further developments on sample complexity on private simple hypothesis selection include [PJL24, PAJL25] under the local model of DP. Other examples within the broader topic of private hypothesis testing include [CDK17, GR18, ADR18, ASZ18, ADKR19, ACFT19, AJM20, CKM$^+$20, BB20].

Asi et al. [ADH$^+$24] give an instance-optimal algorithm for hypothesis selection when the hypothesis class $\mathcal{H}$ contains the true distribution $P$ (i.e. $\mathrm{OPT} = 0$). Their algorithm has logarithmic sample complexity and time complexity $O(n^2 s)$.

## A.1  Industrial applications of DP

We highlight several lines of research in differentially private data analysis motivated by real-world challenges. One example is the partition selection problem, where users aim to compute aggregate statistics over data grouped according to user-specified criteria. To ensure privacy, designers must bound the sensitivity of the statistics, decide which data partition to release, and maintain computational efficiency. Relevant works include Desfontaines et al. [DVG20] and Google's *Plume* system [AGJ$^+$22].

Another direction concerns private analytics of user actions, where the goal is to prevent attackers from learning a user's past behavior by repeated observation of public analytics. LinkedIn's *PriPeARL* [KT18] provides such protection, even towards persistent attackers who observe the analytics overtime.

Privacy leakages also emerge from releasing models trained on sensitive data. To mitigate this problem, Lécuyer et al. [LSV$^+$19] developed *Sage*, a machine learning platform that distributes training across data blocks, monitors privacy loss per blocks, and retires blocks once their privacy budget is depleted.

Lastly, the problem of answering queries across multiple private databases has also been studied. A representative system is *DJoin* by Narayan et al. [NH12].

# B   Proof of Privacy of Algorithm 1

**Lemma B.1.** *Algorithm 1 is $\epsilon$-differentially private.*

*Proof.* In each iteration, Algorithm 1 samples $\mathcal{K}$, a hypothesis list of size $k$, where each hypothesis $H_j$ is drawn with probability proportional to its proxy distance, $\hat{W}_A(H_j)$. In Lemma E.3, we show that each proxy distance has sensitivity $1/s$. We privatize this sampling of hypotheses using the exponential mechanism (Definition 1.4).

After drawing $\mathcal{K}$, the algorithm calculates $\mathsf{score}_{\eta,\mathcal{K},D}(H_j)$ for each hypothesis $H_j$, and searches for a hypothesis with a high $\mathsf{score}_{\eta,\mathcal{K},D}(H_j)$. For each $H_j$, recall that $\mathsf{score}_{\eta,\mathcal{K},D}(H_j)$ is an empirical quantile of the promptingness of $H_j$ on $\mathcal{K}$. $\mathcal{K}$ is treated as publicly available when computing each score. This fact, along with the choice to use quantiles to calculate each score, leads to a low sensitivity of the score function. Specifically, in Lemma E.4, we prove that the score function has sensitivity $2/s$. We privately select a hypothesis with a sufficiently high score using the sparse vector technique (Algorithm 3), allowing us to incur a loss of privacy independent of the number of hypotheses.

All remaining steps in the round either post-process $\mathcal{K}$ and the chosen prompting hypothesis $H_{i_t}$, or reveal no further information about the dataset. Therefore, in each round, we only consider the privacy loss of drawing $s$ hypotheses and finding a prompting hypothesis. By basic composition (Fact 1.7), sampling $k$ hypotheses from $Q$ and calling FIND-PROMPTING-HYPOTHESIS once gives the following privacy loss in each round:

$$\epsilon_2 + k \cdot \epsilon_1 = \frac{\epsilon}{2T} + \frac{\epsilon \cdot k}{2(kT + 1)}.$$

Algorithm 1 takes at most $T$ iterations. The composition of $T$ rounds of the above procedure results in a privacy loss of:

$$T \cdot \left( \frac{\epsilon}{2T} + \frac{\epsilon \cdot k}{2(kT + 1)} \right) = \frac{\epsilon}{2} + \frac{\epsilon \cdot kT}{2(kT + 1)}.$$

Finally, Algorithm 1 samples from $Q$ to obtain output distribution in the last round. Hence, Algorithm 1 has a total privacy loss of:

$$\frac{\epsilon}{2} + \frac{\epsilon \cdot kT}{2(kT + 1)} + \epsilon_1 = \frac{\epsilon}{2} + \frac{\epsilon \cdot kT}{2(kT + 1)} + \frac{\epsilon}{2(kT + 1)} = \epsilon.$$

$\square$

# C   Proof of Correctness of Algorithm 1

The correctness proof of Algorithm 1 proceeds by first showing that, with high probability, three key events occur: ***i)*** the empirical semi-distances are accurate, ***ii)*** in each round, the score of every hypothesis accurately reflects its ability to lift many hypotheses, and ***iii)*** in each round, FIND-PROMPTING-HYPOTHESIS either outputs a high-scoring hypothesis or correctly identifies that no such hypothesis remains. We then show that if these events hold, the algorithm will eventually fail to find a prompting hypothesis and will halt after less than $T$ rounds. Finally, we prove that if the key events hold and the algorithm halts early, the output is, with high probability, less than $(3\mathrm{OPT} + \sigma)$-far from $P$.

### C.1 Key events occur

Let $\sigma_1 = \sigma/4$ and $\sigma_2 = \sigma/4$. The correctness of Algorithm 1 relies on the following key events:

1. With the $s$ samples that we draw from $P$, we calculate the empirical semi-distances between all pairs of hypotheses to within an $\sigma_1$-additive error.

2. In each round of the algorithm, if the score of a hypothesis is at least $\sigma_2/2$, then that hypothesis is $(\sigma_2/2, \eta/4)$-prompting with respect to $Q$. If the score of a hypothesis is less than $\sigma_2$, then that hypothesis is *not* $(\sigma_2, \eta)$-prompting with respect to $Q$.

3. In each round of the algorithm, if FIND-PROMPTING-HYPOTHESIS outputs a hypothesis, then the score of that hypothesis is at least $\sigma_2/2$. If it does not output a hypothesis, then no hypothesis had a score greater than $\sigma_2$.

Each of these events fails to occur with low probability, provided $s$, the number of samples, and $k$, the number of hypotheses sampled at each round, are sufficiently large. We recall the specific assumptions on $s$ and $k$ that guarantee each event as follows:

**Empirical semi-distances are $\sigma_1$-accurate:** Lemma 2.1 ensures that, with probability at least $1 - \beta/6$, the empirical semi-distances are accurate up to an additive factor of $\sigma_1$ if

$$s \geq \frac{1}{2\sigma_1^2} \log(12n/\beta) . \tag{4}$$

**Scores reflect prompting ability:** In Lemma E.5, we show that, in each round, with high probability, the score of each hypothesis reflects whether that hypothesis is prompting with respect to the hypotheses in $Q$. Specifically, the lemma ensures that, with probability at least $1 - \beta/(6T)$, if

$$k \geq \frac{12 \log(6nT/\beta)}{\eta}, \tag{5}$$

then, in round $t$ for all $H_i$, if the score of $H_i$ is at least $\sigma_2/2$, $H_i$ is $(\sigma_2/2, \eta/4)$-prompting with respect to $Q$, and, if the score of $H_i$ is less than $\sigma_2$, $H_i$ is *not* $(\sigma_2, \eta)$-prompting with respect to $Q$.

**FIND-PROMPTING-HYPOTHESIS succeeds:** In Theorem 6, we show that, in each round, with high probability, the FIND-PROMPTING-HYPOTHESIS procedure succeeds in either finding a prompting hypothesis or identifying that there are no such hypotheses. Specifically, the theorem ensures that, with probability at least $1 - \beta/(6T)$, if

$$\Delta \leq \frac{\sigma_2 \epsilon_2}{32 \log(12nT/\beta)} , \tag{6}$$

then, in round $t$, if FIND-PROMPTING-HYPOTHESIS$(\epsilon_2, \Delta = 2/s, \sigma_2, \eta, \mathcal{H} \setminus A, \mathcal{K})$ outputs $H_{i_t}$, the score of $H_{i_t}$ is at least $\sigma_2/2$, and, if the procedure outputs $\bot$, each $H_{i_t}$ has a score less than $\sigma_2$. Note that this bound on $\Delta$ implies the following bound on $s$, as $\Delta = 2/s$:

$$s \geq \frac{64}{\sigma_2 \epsilon_2} \log(12nT/\beta) , \tag{7}$$

For now, assume that $s, T$ and $k$ satisfy these requirements. In Section C.5, we will give a precise choice of parameters that satisfies these, along with other constraints. Then, the probability that at

least one of the three key events does not occur is:

$\mathbf{Pr}[\text{at least one key event fails}] \leq \mathbf{Pr}[\text{semi-distance estimation fails}]$

$$+ \sum_{t=1}^{T} \mathbf{Pr}[\text{at least one score is inaccurate in round } t]$$

$$+ \sum_{t=1}^{T} \mathbf{Pr}[\text{FIND-PROMPTING-HYPOTHESIS fails in round } t]$$

$$\leq \frac{\beta}{6} + \sum_{t=1}^{T} \frac{\beta}{6T} + \sum_{t=1}^{T} \frac{\beta}{6T}$$

$$= \frac{\beta}{2} \ .$$

## C.2 Algorithm halts early

If FIND-PROMPTING-HYPOTHESIS outputs a hypothesis with score at least $\sigma_2/2$, then that hypothesis must be able to $\sigma_2/2$-lift at least an $\eta/2$-fraction of $\mathcal{K}$. If its scores are an accurate representation of its ability to lift $Q$, then $H_i$ is $(\sigma_2/2, \eta/4)$-prompting over $Q$. In this section, we show that, if each hypothesis added to the prompting set is truly $(\sigma_2/2, \eta/4)$-prompting–that is, if the second and third events described in the previous section hold–, then the algorithm will reach Line 16 and halt before executing $T$ rounds.

**Theorem 4.** *Let $\sigma' = \sigma_2/2, \eta' = \eta/4$. Assume $\exp\left(-\frac{\epsilon_1 \sigma'}{2\Delta(\hat{W}_A)}\right) < \frac{1}{2}$. Further, assume that each hypothesis added to the prompting set is $(\sigma', \eta')$-prompting with respect to $Q$ in the round it is added. Then Algorithm 1 terminates after at most $\min\left(\frac{1}{\log\left(1 + \frac{\eta'}{2}\right)}\left(\log(n) + \frac{\epsilon_1}{2\Delta(\hat{W}_A)} \cdot OPT\right), n\right)$ rounds.*

**Proof Sketch** Initially, $Q$ is the uniform distribution over $\{H_1, \ldots, H_n\}$, as each proxy distance $\hat{W}_A^{(t)}(H_j)$ is initialized to 0. After each round of the algorithm, a new hypothesis $H_{i_t}$ is added to the prompting set and the proxy distance of each hypothesis either increases or remains constant. If each $H_{i_t}$ is truly prompting over $Q$, the normalization term of the exponential mechanism decreases at each round, amplifying the probabilities of the hypotheses with proxy distances that are relatively unchanged by the addition of $H_t$. By characterizing these changes, we give an upper bound $\tilde{T}$ on the number of rounds our algorithm will execute before returning a hypothesis. Note that this upper bound must be less than $n$, as every hypothesis is added to the prompting set at most once. If we choose $T$ to be greater than $\tilde{T}$, the algorithm will halt before executing $T$ rounds.

To formally prove Theorem 4, we introduce the following notation, exactly describing the probability distribution induced by the exponential mechanism in each round.

**Definition C.1** (Exponential mechanism at round $t$). *Recall that, at round $t$, the hypothesis sampling distribution, $Q^{(t)}$, is defined as follows:*

$$Q^{(t)}(H_j) = \frac{\exp\left(-\frac{\epsilon_1}{2\Delta(\hat{W}_A)} \cdot \hat{W}_A^{(t)}(H_j)\right)}{Z^{(t)}} \ ,$$

*where*

$$Z^{(t)} = \sum_{j=1}^{n} \exp\left(-\frac{\epsilon_1}{2\Delta(\hat{W}_A)} \cdot \hat{W}_A^{(t)}(H_j)\right) \ , \tag{8}$$

*and $\hat{W}_A^{(t)}(H)$ is the proxy distance of $H$ at round $t$.*

We also require the following lemma, which demonstrates that $Z^{(\ell)}$ decreases with each round, and bounds the amount of this decrease.

**Lemma C.2.** *Let $\ell \in \{1, 2, \ldots\}$. Let $\sigma', \eta' \in (0, 1]$. Assume $\exp\left(-\frac{\epsilon_1 \sigma'}{2\Delta(\hat{W}_A)}\right) < \frac{1}{2}$. Define $Z^{(\ell)}$ and $Z^{(\ell+1)}$ as in Definition 8. Assume the hypothesis $H_{i_\ell}$ added to the prompting set in round $\ell$ is $(\sigma', \eta')$-prompting with respect to $Q^{(\ell)}$. Then, we have:*

$$\frac{Z^{(\ell+1)}}{Z^{(\ell)}} \leq 1 - \frac{\eta'}{2} . \tag{9}$$

Before proving Lemma C.2, we introduce, for each hypothesis $H_j$, a value $u^{(\ell+1)}(H_j)$ describing the increase in $H_j$'s proxy distance between rounds $\ell$ and $\ell + 1$:

$$u^{(\ell+1)}(H_j) := \hat{W}_A^{(\ell+1)}(H_j) - \hat{W}_A^{(\ell)}(H_j). \tag{10}$$

Recall that we define $\hat{W}_A^{(\ell+1)}(H_j)$ to be the maximum empirical semi-distance between $H_j$ and the hypotheses in the prompting set $A$. Thus, when we add $H_{i_\ell}$ to the prompting set in round $\ell$, $\hat{W}_A^{(\ell+1)}(H_j)$ is the maximum of $\hat{w}_{i_\ell}(H_j)$ and $\hat{W}_A^{(\ell)}(H_j)$:

$$\hat{W}_A^{(\ell+1)}(H_j) = \max\left\{ \hat{w}_{i_\ell}(H_j), \hat{W}_A^{(\ell)}(H_j) \right\}.$$

The difference between $\hat{W}_A^{(\ell+1)}(H_j)$ and $\hat{W}_A^{(\ell)}(H_j)$ is thus lower bounded by the difference between $\hat{W}_A^{(\ell+1)}(H_j)$ and $\hat{w}_{i_\ell}(H_j)$:

$$u^{(\ell+1)}(H_j) = \hat{W}_A^{(\ell+1)}(H_j) - \hat{W}_A^{(\ell)}(H_j) \geq \hat{w}_{i_\ell}(H_j) - \hat{W}_A^{(\ell)}(H_j).$$

If $H_\ell$ is $(\sigma', \eta')$-prompting, the definition of prompting allows us to lower bound the probability that this difference is greater than $\sigma'$:

$$\mathbf{Pr}_{H_j \sim Q^{(\ell)}}\left[ \hat{w}_{i_\ell}(H_j) - \hat{W}_A^{(\ell)}(H_j) \geq \sigma' \right] \geq \eta' \implies \mathbf{Pr}_{H_j \sim Q^{(\ell)}}\left[ u^{(\ell+1)}(H_j) \geq \sigma' \right] \geq \eta'. \tag{11}$$

To prove Lemma C.2, we relate the ratio in question to the expected value of a function of these differences. If the hypothesis added to the prompting set at round $\ell$ is prompting, Equation 11 gives a tail bound on each of these differences. Combining this expected value and tail bound with careful consideration yields the desired bound.

*Proof of Lemma C.2.* Considering the ratio in question, we expand $Z^{(\ell+1)}$ as follows:

$$\frac{Z^{(\ell+1)}}{Z^{(\ell)}} = \frac{1}{Z^{(\ell)}} \sum_{j=1}^{n} \exp\left( -\frac{\epsilon_1}{2\Delta(\hat{W}_A)} \cdot \hat{W}_A^{(\ell+1)}(H_j) \right). \tag{12}$$

Rewriting $\hat{W}_A^{(\ell+1)}(H_j)$ in terms of $u^{(\ell+1)}(H_j)$ and $\hat{W}_A^{(\ell)}(H_j)$ for every $j$, we have:

$$\frac{Z^{(\ell+1)}}{Z^{(\ell)}} = \frac{1}{Z^{(\ell)}} \sum_{j=1}^{n} \exp\left( -\frac{\epsilon_1}{2\Delta(\hat{W}_A)} \cdot \left( \hat{W}_A^{(\ell)}(H_j) + u^{(\ell+1)}(H_j) \right) \right)$$

$$= \sum_{j=1}^{n} \frac{1}{Z^{(\ell)}} \exp\left( -\frac{\epsilon_1}{2\Delta(\hat{W}_A)} \cdot \hat{W}_A^{(\ell)}(H_j) \right) \cdot \exp\left( -\frac{\epsilon_1}{2\Delta(\hat{W}_A)} \cdot u^{(\ell+1)}(H_j) \right)$$

$$= \sum_{j=1}^{n} Q^{(\ell)}(H_j) \cdot \exp\left( -\frac{\epsilon_1}{2\Delta(\hat{W}_A)} \cdot u^{(\ell+1)}(H_j) \right).$$

The second equality follows from the definition of $Q^{(\ell)}(H_j)$. Then, notice that the final line above forms an expectation over $Q^{(\ell)}$. We can rewrite it as such:

$$= \mathbf{E}_{H_j \sim Q^{(\ell)}}\left[ \exp\left( -\frac{\epsilon_1}{2\Delta(\hat{W}_A)} \cdot u^{(\ell+1)}(H_j) \right) \right]. \tag{13}$$

Assuming the hypothesis added to the prompting set at round $\ell$ is prompting, Equation 11 applies. We will use that bound on the probability to bound the expectation in Equation 13. To do so, as the argument of the expectation is non-negative, we begin by applying the integral identity [Ver18]:

$$
\mathbf{E}_{H_j \sim Q^{(\ell)}} \left[ \exp \left( -\frac{\epsilon_1}{2\Delta(\hat{W}_A)} \cdot u^{(\ell+1)}(H_j) \right) \right]
$$
$$
= \int_0^\infty \mathbf{Pr}_{H_j \sim Q^{(\ell)}} \left[ \exp \left( -\frac{\epsilon_1}{2\Delta(\hat{W}_A)} \cdot u^{(\ell+1)}(H_j) \right) > t \right] dt.
$$

Rearranging to isolate $u^{(\ell+1)}(H_j)$ within the probability gives:

$$
= \int_0^\infty \mathbf{Pr}_{H_j \sim Q^{(\ell)}} \left[ u^{(\ell+1)}(H_j) < \frac{2\Delta(\hat{W}_A)}{\epsilon_1} \log \frac{1}{t} \right] dt.
$$

We apply a change of variables, letting $v = \frac{2\Delta(\hat{W}_A)}{\epsilon_1} \log \frac{1}{t}$:

$$
= \int_\infty^{-\infty} \mathbf{Pr}_{H_j \sim Q^{(\ell)}} \left[ u^{(\ell+1)}(H_j) < v \right] \cdot \left( -\frac{\epsilon_1}{2\Delta(\hat{W}_A)} \right) \cdot \exp \left( -\frac{v\epsilon_1}{2\Delta(\hat{W}_A)} \right) dv
$$
$$
= \int_{-\infty}^{\infty} \mathbf{Pr}_{H_j \sim Q^{(\ell)}} \left[ u^{(\ell+1)}(H_j) < v \right] \cdot \left( \frac{\epsilon_1}{2\Delta(\hat{W}_A)} \right) \cdot \exp \left( -\frac{v\epsilon_1}{2\Delta(\hat{W}_A)} \right) dv.
$$

Because $u^{(\ell+1)}(H_j)$ is non-negative, the integrand is 0 when $v < 0$, and we need only to evaluate the integral from $v = 0$ to $\infty$:

$$
= \int_0^\infty \mathbf{Pr}_{H_j \sim Q^{(\ell)}} \left[ u^{(\ell+1)}(H_j) < v \right] \cdot \left( \frac{\epsilon_1}{2\Delta(\hat{W}_A)} \right) \cdot \exp \left( -\frac{v\epsilon_1}{2\Delta(\hat{W}_A)} \right) dv.
$$

By Equation 11, we know that, for any $v \leq \sigma'$, $\mathbf{Pr}\left[ u^{(\ell+1)}(H_j) \geq v \right] \geq \eta'$. This implies that, for any $v \in [0, \sigma']$, we can *upper bound* $\mathbf{Pr}\left[ u^{(\ell+1)}(H_j) < v \right]$ by $1 - \eta'$. At the same time, for $v > \sigma'$, we can upper bound $\mathbf{Pr}\left[ u^{(\ell+1)}(H_j) < v \right]$ by 1. Applying these bounds, the resulting integral is:

$$
\leq \int_0^{\sigma'} (1 - \eta') \cdot \left( \frac{\epsilon_1}{2\Delta(\hat{W}_A)} \right) \cdot \exp \left( -\frac{v\epsilon_1}{2\Delta(\hat{W}_A)} \right) dv
$$
$$
+ \int_{\sigma'}^\infty 1 \cdot \left( \frac{\epsilon_1}{2\Delta(\hat{W}_A)} \right) \cdot \exp \left( -\frac{v\epsilon_1}{2\Delta(\hat{W}_A)} \right) dv
$$
$$
= \int_0^\infty 1 \cdot \left( \frac{\epsilon_1}{2\Delta(\hat{W}_A)} \right) \cdot \exp \left( -\frac{v\epsilon_1}{2\Delta(\hat{W}_A)} \right) dv - \int_0^{\sigma'} \eta' \cdot \left( \frac{\epsilon_1}{2\Delta(\hat{W}_A)} \right) \cdot \exp \left( -v \cdot \frac{\epsilon_1}{2\Delta(\hat{W}_A)} \right) dv
$$
$$
= \lim_{t \to \infty} -\exp \left( -v \cdot \frac{\epsilon_1}{2\Delta(\hat{W}_A)} \right) \Big|_{v=0}^{t} + \eta' \exp \left( -v \cdot \frac{\epsilon_1}{2\Delta(\hat{W}_A)} \right) \Big|_{v=0}^{\sigma'}
$$
$$
= \lim_{t \to \infty} \exp \left( -t \cdot \frac{\epsilon_1}{2\Delta(\hat{W}_A)} \right) + \exp \left( -0 \cdot \frac{\epsilon_1}{2\Delta(\hat{W}_A)} \right)
$$
$$
+ \eta' \left( \exp \left( -\sigma' \cdot \frac{\epsilon_1}{2\Delta(\hat{W}_A)} \right) - \exp \left( -0 \cdot \frac{\epsilon_1}{2\Delta(\hat{W}_A)} \right) \right)
$$
$$
= 1 + \eta' \left( \exp \left( -\sigma' \cdot \frac{\epsilon_1}{2\Delta(\hat{W}_A)} \right) - 1 \right).
$$

Under the assumption $\exp\left(-\frac{\epsilon_1 \sigma'}{2\Delta(\hat{W}_A)}\right) < \frac{1}{2}$, the following holds:

$$1 - \eta'\left(1 - \exp\left(-\frac{\epsilon_1 \sigma'}{2\Delta(\hat{W}_A)}\right)\right) \leq 1 - \frac{\eta'}{2}.$$

$\square$

Lemma C.2 implies that when hypothesis $H_{i_\ell}$ is added to the prompting set, the probability of sampling a hypothesis $H_j$ which is not significantly lifted by $H_{i_\ell}$ increases. As we continue to add prompting hypotheses which do not significantly lift $H_j$, $Q(H_j)$ continues to increase. As this probability approaches 1 and $H_j$ is repeatedly sampled by the exponential mechanism, either we will find a hypothesis which lifts $H_j$, in which case its probability will decrease, or we will halt, as we could not find such a hypothesis. At any given round, we know that the probability of sampling any hypothesis cannot grow until it exceeds 1. We use this fact to upper bound the number of rounds of Algorithm 1, thus proving Theorem 4.

*Proof of Theorem 4.* Note that the algorithm cannot add more than $n$ hypotheses to the set $A$, as it will only check for a prompting hypothesis among the hypotheses in $\mathcal{H} \setminus A$. As a result, the algorithm does not run for more than $n$ rounds.

Fix $t \geq 0$. Consider the probability that $H_j$ is selected by the exponential mechanism at round $t+1$, $Q^{(t+1)}(H_j)$:

$$
\begin{aligned}
Q^{(t+1)}(H_j) &= \frac{1}{Z^{(t+1)}} \cdot \exp\left(-\frac{\epsilon_1}{2\Delta(\hat{W}_A)} \cdot \hat{W}_A^{(t+1)}(H_j)\right) \\
&= \frac{1}{Z^{(t+1)}} \cdot \exp\left(-\frac{\epsilon_1}{2\Delta(\hat{W}_A)} \cdot \left(\hat{W}_A^{(t)}(H_j) + u^{(t+1)}(H_j)\right)\right) \\
&= \frac{Z^{(t)}}{Z^{(t+1)}} \cdot \frac{1}{Z^{(t)}} \cdot \exp\left(-\frac{\epsilon_1}{2\Delta(\hat{W}_A)} \cdot \hat{W}_A^{(t)}(H_j)\right) \cdot \exp\left(-\frac{\epsilon_1}{2\Delta(\hat{W}_A)} \cdot u^{(t+1)}(H_j)\right) \\
&= \frac{Z^{(t)}}{Z^{(t+1)}} \cdot Q^{(t)}(H_j) \cdot \exp\left(-\frac{\epsilon_1}{2\Delta(\hat{W}_A)} \cdot u^{(t+1)}(H_j)\right).
\end{aligned}
$$

Unfolding this expression over all rounds prior to $t$, we obtain:

$$Q^{(t)}(H_j) = \left(\prod_{\ell=1}^{t} \frac{Z^{(\ell-1)}}{Z^{(\ell)}}\right) \cdot Q^{(0)}(H_j) \cdot \exp\left(-\frac{\epsilon_1}{2\Delta(\hat{W}_A)} \cdot \sum_{\ell=1}^{t} u^{(\ell)}(H_j)\right). \qquad (14)$$

Recall that $Q^{(0)}$ follows a uniform distribution, implying $Q^{(0)}(H_j) = \frac{1}{n}$. Further, as $\hat{W}_A^{(0)}(H_j) = 0$ and $\hat{W}_A^{(t)}(H_j) = \hat{W}_A^{(0)}(H_j) + \sum_{\ell=1}^{t} u^{(\ell)}(H_j)$, we have $\hat{W}_A^{(t)}(H_j) = \sum_{\ell=1}^{t} u^{(\ell)}(H_j)$. As a result, we can simplify Equation 14 to:

$$Q^{(t)}(H_j) = \left(\prod_{\ell=1}^{t} \frac{Z^{(\ell-1)}}{Z^{(\ell)}}\right) \cdot \frac{1}{n} \cdot \exp\left(-\frac{\epsilon_1}{2\Delta(\hat{W}_A)} \cdot \hat{W}_A^{(t)}(H_j)\right). \qquad (15)$$

By Lemma C.2, we know $\frac{Z^{(\ell+1)}}{Z^{(\ell)}} \leq 1 - \frac{\eta'}{2}$ for all $\ell$. Consequently, $\frac{Z^{(\ell-1)}}{Z^{(\ell)}} \geq \frac{1}{1-\frac{\eta'}{2}} \geq 1 + \frac{\eta'}{2}$. Additionally, $\hat{W}_A^{(\ell)}(H_j)$ is upper bounded by $W(H_j)$. Combining these two bounds and Equation 15, we have:

$$Q^{(t)}(H_{i^*}) \geq \left(1 + \frac{\eta'}{2}\right)^t \cdot \frac{1}{n} \cdot \exp\left(-\frac{\epsilon_1}{2\Delta(\hat{W}_A)} \cdot W(H_j)\right). \qquad (16)$$

The above expression of $Q^{(t)}(H_j)$ holds for all $H_j$–including $H_{i^*}$. If we focus specifically on $H_{i^*}$, we know $W(H_{i^*}) \leq \text{OPT}$, leading to the bound:

$$Q^{(t)}(H_{i^*}) \geq \left(1 + \frac{\eta'}{2}\right)^t \cdot \frac{1}{n} \cdot \exp\left(-\frac{\epsilon_1}{2\Delta(\hat{W}_A)} \cdot \text{OPT}\right) . \tag{17}$$

Because $Q^{(t)}(H_{i^*})$ represents a probability, it is upper bounded by 1. The lower bound given in Equation 17 is thus also upper bounded by 1. We use these bounds to establish an upper bound on $t$:

$$1 \geq \left(1 + \frac{\eta'}{2}\right)^t \cdot \frac{1}{n} \cdot \exp\left(-\frac{\epsilon_1}{2\Delta(\hat{W}_A)} \cdot \text{OPT}\right) \implies n \cdot \exp\left(\frac{\epsilon_1}{2\Delta(\hat{W}_A)} \cdot \text{OPT}\right) \geq \left(1 + \frac{\eta'}{2}\right)^t$$

$$\implies \log(n) + \frac{\epsilon_1}{2\Delta(\hat{W}_A)} \cdot \text{OPT} \geq t \log\left(1 + \frac{\eta'}{2}\right)$$

$$\implies \frac{\log(n) + \frac{\epsilon_1}{2\Delta(\hat{W}_A)} \cdot \text{OPT}}{\log\left(1 + \frac{\eta'}{2}\right)} \geq t .$$

Therefore, our algorithm will terminate after at most $\frac{\log(n) + \frac{\epsilon_1}{2\Delta(\hat{W}_A)} \cdot \text{OPT}}{\log\left(1 + \frac{\eta'}{2}\right)}$ rounds.

$\square$

### C.3 Algorithm outputs a valid hypothesis

**Theorem 5.** *Assume $\sigma \geq \frac{8\Delta(\hat{W}_A)}{\epsilon_1} \log(4n/\beta)$. Assume the key events hold and Algorithm 1 reaches Line 16 and outputs $\hat{H}$. Then, with probability at most $\beta/2$, $\|\hat{H} - P\|_{TV} > 3 \cdot \text{OPT} + \sigma$.*

*Proof.* If Algorithm 1 breaks in round $t$, it must be that FIND-PROMPTING-HYPOTHESIS was unable to find a prompting hypothesis and returned $\perp$. Under the assumption that FIND-PROMPTING-HYPOTHESIS succeeds, this implies that, at round $t$, all hypotheses had scores less than $\sigma_2$, implying that they were unable to $\sigma_2$-lift more than an $\eta/2$ fraction of $\mathcal{K}$. We've additionally assumed that the score of each hypothesis reflects its promptingness. That is, if the score of a hypothesis is less than $\sigma_2$, it is not $(\sigma_2, \eta)$-prompting. Therefore, under our assumptions, no $H_i \in \mathcal{H}$ is $(\sigma_2, \eta)$-prompting. Specifically, $H_{i^*}$ is not $(\sigma_2, \eta)$-prompting. This guarantees that:

$$\mathbf{Pr}_{\hat{H} \sim Q}\left[\hat{w}_{i^*}(\hat{H}) - \hat{W}_A(\hat{H}) \geq \sigma_2\right] < \eta . \tag{18}$$

We also know, by the utility guarantee of the exponential mechanism given in Lemma 1.6, that, with probability at least $1 - \beta/4$, for $\hat{H} \sim Q$:

$$\hat{W}_A(\hat{H}) \leq \min_{i \in [n]} \hat{W}_A(H_i) + \frac{2\Delta(\hat{W}_A)}{\epsilon_1} \log(n/4\beta) . \tag{19}$$

We can bound $\min_{i \in [n]} \hat{W}_A(H_i)$ as follows:

$$\min_{i \in [n]} \hat{W}_A(H_i) \leq \hat{W}_A(H_{i^*}) = \max_{H_j \in A} \hat{w}_j(H_{i^*}) \leq \max_{j \in [n]} \hat{w}_j(H_{i^*}) . \tag{20}$$

Recall that we have assumed that the empirical semi-distance estimates are accurate up to an additive factor of $\sigma_1$. This implies:

$$\min_{i \in [n]} \hat{W}_A(H_i) \leq \max_{j \in [n]} \hat{w}_j(H_{i^*}) \leq \max_{j \in [n]} w_j(H_{i^*}) + \sigma_1 \leq \|H_{i^*} - P\|_{TV} + \sigma_1 = \text{OPT} + \sigma_1 . \tag{21}$$

Combining Equation 19, and Equation 21, we have, with probability at least $1 - \beta/4$, the following bound on, $\hat{W}_A(\hat{H})$, the proxy distance of our outputted hypothesis:

$$\hat{W}_A(\hat{H}) \leq \text{OPT} + \sigma_1 + \frac{2\Delta(\hat{W}_A)}{\epsilon_1} \log\left(4n/\beta\right) . \tag{22}$$

Because $\hat{H}$ is drawn from $Q$, we know, from Equation 18, that, with probability at least $1 - \eta$, we can bound the empirical semi-distance $\hat{w}_{i^*}(\hat{H})$ as:

$$\hat{w}_{i^*}(\hat{H}) < \hat{W}_A(\hat{H}) + \sigma_2 . \tag{23}$$

Combining Equation 22 and Equation 23, we have, with probability at least $1 - (\beta/4 + \eta)$, the following bound on $\hat{w}_{i^*}(\hat{H})$:

$$\hat{w}_{i^*}(\hat{H}) < \text{OPT} + \sigma_2 + \frac{2\Delta(\hat{W}_A)}{\epsilon_1} \log\left(4n/\beta\right) + \sigma_1 . \tag{24}$$

By Equation 3, we have $\|\hat{H} - P\|_{\text{TV}} \leq 2\text{OPT} + w_{i^*}(\hat{H})$. Together with Equation 24, this yields:

$$\begin{aligned}
\|\hat{H} - P\|_{\text{TV}} &\leq 2\text{OPT} + w_{i^*}(\hat{H}) \\
&\leq 2\text{OPT} + \hat{w}_{i^*}(\hat{H}) + \sigma_1 \\
&< 3\text{OPT} + \sigma_2 + \frac{2\Delta(\hat{W}_A)}{\epsilon_1} \log\left(4n/\beta\right) + 2\sigma_1 \\
&\leq 3\text{OPT} + \sigma/4 + \sigma/4 + 2 \cdot \sigma/4 \\
&= 3\text{OPT} + \sigma .
\end{aligned}$$

The second-to-last inequality holds because we set $\sigma_2 = \sigma/4, \sigma_1 = \sigma/4$, and we assume $\sigma \geq \frac{8\Delta(\hat{W}_A)}{\epsilon_1} \log\left(4n/\beta\right)$. Recall that our algorithm sets $\eta = \beta/4$. Ultimately, we have shown:

$$\mathbf{Pr}_{\hat{H} \sim Q}\left[\|\hat{H} - P\|_{\text{TV}} > 3 \cdot \text{OPT} + \sigma \mid \text{algorithm breaks and key events occur}\right] \leq \eta + \beta/4 = \beta/2 .$$

$\square$

## C.4 Overall correctness

We have shown, under certain constraints on $s, T$ and $k$, that the probability that key events do not all hold is at most $\beta/2$, and that, if the key events do hold and the algorithm halts before $T$ rounds, then the probability that the algorithm outputs a far hypothesis $\hat{H}$ is at most $\beta/2$.

Hence, if $s, T$ and $k$ satisfy the required constraints, we can bound the probability that the algorithm outputs a hypothesis $\hat{H}$ greater than $(3\text{OPT} + \sigma)$-far as:

$$\begin{aligned}
&\mathbf{Pr}\left[\|\hat{H} - P\|_{\text{TV}} > 3 \cdot \text{OPT} + \sigma\right] \\
&= \mathbf{Pr}\left[\|\hat{H} - P\|_{\text{TV}} > 3 \cdot \text{OPT} + \sigma \mid \text{key events occur}\right] \mathbf{Pr}[\text{key events occur}] \\
&\quad + \mathbf{Pr}\left[\|\hat{H} - P\|_{\text{TV}} > 3 \cdot \text{OPT} + \sigma \mid \textit{not} \text{ key events occur}\right] \mathbf{Pr}[\textit{not} \text{ key events occur}] \\
&\leq \beta/2 \cdot 1 + 1 \cdot \beta/2 \\
&= \beta .
\end{aligned}$$

## C.5 Sample complexity

In this section, we give exact settings of $s, T,$ and $k$ that satisfy the constraints given throughout our proof of correctness.

Recall that, throughout our proof of correctness, we make the following assumptions:

1. To accurately estimate semi-distances:

$$s \geq \frac{1}{2\sigma_1^2} \log(12n/\beta) \tag{25}$$

2. To accurately approximate prompting-ness:

$$k \geq \frac{12 \log(6nT/\beta)}{\eta} \tag{26}$$

3. For FIND-PROMPTING-HYPOTHESIS to succeed:

$$s \geq \frac{64}{\sigma_2 \epsilon_2} \log(12nT/\beta) \tag{27}$$

4. To ensure the final output has a low proxy distance:

$$\sigma \geq \frac{8}{\Delta(\hat{W}_A)} \epsilon_1 \log(4n/\beta) \tag{28}$$

5. To bound the number of rounds the algorithm will execute:

$$\exp\left(-\frac{\epsilon_1 \sigma'}{2\Delta(\hat{W}_A)}\right) < \frac{1}{2} \tag{29}$$

6. To ensure the algorithm does not halt prematurely:

$$T \geq \frac{1}{\log(1 + \eta'/2)}\left(\log n + \frac{\epsilon_1}{2\Delta(\hat{W}_A)}\mathrm{OPT}\right). \tag{30}$$

Throughout the algorithm and analysis, we also have the following parameter settings:

$$\sigma_1 = \tfrac{\sigma}{4}, \quad \sigma_2 = \tfrac{\sigma}{4}, \quad \eta = \tfrac{\beta}{4}, \quad \eta' = \tfrac{\eta}{4}, \quad \sigma' = \tfrac{\sigma_2}{2},$$

$$\epsilon_2 = \tfrac{\epsilon}{2T}, \quad \epsilon_1 = \tfrac{\epsilon}{2(kT+1)}, \quad \Delta(\hat{W}_A) = \tfrac{1}{s}.$$

Combining these settings with our above assumptions and constraints, we have the following set of requirements for $s, T$ and $k$:

$$s \geq \max\left(\frac{8}{\sigma^2}\log\left(12n/\beta\right), \frac{512T}{\sigma\epsilon}\log\left(12nT/\beta\right), \frac{16\log(2)(kT+1)}{\sigma\epsilon}, \frac{16(kT+1)}{\sigma\epsilon}\log\left(4n/\beta\right)\right),$$

$$T \geq \frac{1}{\log\left(1 + \frac{\beta}{32}\right)}\left(\log n + \frac{\epsilon s}{4(kT+1)}\mathrm{OPT}\right), \text{ and}$$

$$k \geq \frac{48}{\beta}\log\left(6nT/\beta\right).$$

We claim that the following settings of $s, T$, and $k$ will satisfy these requirements:

$$s = \frac{32 \cdot 96 \cdot 33 \cdot 16}{\beta^2 \sigma^2 \epsilon}\log^3\left(6n/\beta\right), \tag{31}$$

$$T = \min\left(\frac{33 \cdot 16}{\beta\sigma}\log\left(6n/\beta\right), n\right), \tag{32}$$

$$k = \frac{96}{\beta}\log\left(6n/\beta\right). \tag{33}$$

**s satisfies constraints**   As both $T$ and $k$ are logarithmic in $n$, the fourth argument in the constraint on $s$ will dominate. For completeness, we show that each argument of this constraint is less than our choice of $s$.

Beginning with the first argument, we have:

$$s = \frac{32 \cdot 96 \cdot 33 \cdot 16}{\beta^2 \sigma^2 \epsilon} \log^3 (6n/\beta) \geq \frac{8}{\sigma^2} \left(\log (6n/\beta) + \log 2\right) = \frac{8}{\sigma^2} \log (12n/\beta) \ .$$

For the second argument, we have:

$$
\begin{aligned}
s &= \frac{32 \cdot 96 \cdot 33 \cdot 16}{\beta^2 \sigma^2 \epsilon} \log^3 (6n/\beta) \\
&= \frac{32}{\sigma \epsilon} \log (6n/\beta) \cdot \frac{96}{\beta} \log (6n/\beta) \cdot \frac{33 \cdot 16}{\beta \sigma} \log (6n/\beta) \\
&= \frac{32 \cdot 96 T}{\beta \sigma \epsilon} \log (6n/\beta) \log (6n/\beta) \\
&\geq \frac{32 \cdot 48 T}{\beta \sigma \epsilon} \left(\log (6n/\beta) + \log 2\right) \log (6n/\beta) \\
&\geq \frac{32 \cdot 48 T}{\beta \sigma \epsilon} \log (12n/\beta) \log (6n/\beta) \ .
\end{aligned}
\tag{34}
$$

Then, because $n \geq T$, we have the following sequence of inequalities, continued from Equation 34:

$$
\begin{aligned}
s &\geq \frac{32 \cdot 24 T}{\beta \sigma \epsilon} \log \left((12n/\beta)^2\right) \log (6n/\beta) \\
&\geq \frac{32 \cdot 24 T}{\beta \sigma \epsilon} \log \left(12n^2/\beta\right) \log (6n/\beta) \\
&\geq \frac{32 \cdot 24 T}{\beta \sigma \epsilon} \log \left(12nT/\beta\right) \log (6n/\beta) \\
&\geq \frac{512 T}{\sigma \epsilon} \log \left(12nT/\beta\right) \ .
\end{aligned}
$$

Note that the third argument is entirely subsumed by the fourth. Therefore, for both the third and fourth argument, we have:

$$
\begin{aligned}
s &= \frac{32 \cdot 96 \cdot 33 \cdot 16}{\beta^2 \sigma^2 \epsilon} \log^3 (6n/\beta) \\
&= \frac{32}{\sigma \epsilon} \log (6n/\beta) \cdot \frac{96}{\beta} \log (6n/\beta) \cdot \frac{33 \cdot 16}{\beta \sigma} \log (6n/\beta) \\
&\geq \frac{32 k T}{\sigma \epsilon} \log (6n/\beta) \\
&\geq \frac{16(kT + 1)}{\sigma \epsilon} \log(4n/\beta) \\
&\geq \frac{16 \log(2)(kT + 1)}{\sigma \epsilon} \ .
\end{aligned}
$$

**T satisfies constraints**   We now show that our choice of $T$ exceeds the bound established in Theorem 4. First, observe that we can lower bound $T$ by:

$$T = \frac{33 \cdot 16}{\beta \sigma} \log(6n/\beta) \geq \frac{33}{\beta} \left(\log n + \frac{8}{\sigma} \log(6n/\beta)\right) \ . \tag{35}$$

We can then expand the second term of this equation and rewrite it in terms of $s, T$ and $k$, yielding:

$$T \geq \frac{33}{\beta} \left( \log n + \frac{\epsilon}{4} \cdot \frac{32 \cdot 96 \cdot 33 \cdot 16}{\beta^2 \sigma^2 \epsilon} \log^3 (6n/\beta) \cdot \frac{\beta}{96 \log (6n/\beta)} \cdot \frac{\beta \sigma}{33 \cdot 16 \log(6n/\beta)} \right)$$

$$= \frac{33}{\beta} \left( \log n + \frac{\epsilon}{4} \cdot s \cdot \frac{1}{k} \cdot \frac{1}{T} \right)$$

$$= \frac{33}{\beta} \left( \log n + \frac{\epsilon s}{4kT} \right)$$

$$\geq \frac{33}{\beta} \left( \log n + \frac{\epsilon s}{4(kT+1)} \right) \ .$$

Applying the fact that $\beta < 1$ and the inequality $\log(1+x) \geq \frac{x}{1+x}$ for all $x > -1$, we can then show:

$$T \geq \frac{32}{\beta} \left( 1 + \frac{1}{32} \right) \left( \log n + \frac{\epsilon s}{4(kT+1)} \right)$$

$$> \frac{32}{\beta} \left( 1 + \frac{\beta}{32} \right) \left( \log n + \frac{\epsilon s}{4(kT+1)} \right)$$

$$\geq \frac{1 + \beta/32}{\beta/32} \left( \log n + \frac{\epsilon s}{4(kT+1)} \right)$$

$$= \frac{1 + \beta/32}{\beta/32} \left( \log n + \frac{\epsilon s}{4(kT+1)} \right)$$

$$\geq \frac{1}{\log (1 + \beta/32)} \left( \log n + \frac{\epsilon s}{4(kT+1)} \right) \ .$$

Finally, because $\mathrm{OPT} \leq 1$, we know $T$ must be greater than the number of rounds required for the algorithm to terminate.

$$T > \frac{1}{\log (1 + \beta/32)} \left( \log n + \frac{\epsilon s \cdot \mathrm{OPT}}{4(kT+1)} \right) \ .$$

**k satisfies constraints**  Recall that $T$ must be less than $n$, as we do not allow any hypothesis to be added to the prompting set more than once. Then, our choice of $k$ satisfies the constraint as follows:

$$k = \frac{96}{\beta} \log (6n/\beta) = \frac{48}{\beta} \log \left( (6n/\beta)^2 \right) \geq \frac{48}{\beta} \log(6n^2/\beta) \geq \frac{48}{\beta} \log(6nT/\beta) \ .$$

## D   Proof of Time Complexity of Algorithm 1

In this section, we prove the time complexity of Algorithm 1.

**Lemma D.1.** *Algorithm 1 takes* $\Theta \left( \min \left( \frac{1}{\beta^4 \sigma^3 \epsilon} \cdot n \cdot \log^5(n/\beta) , \ \frac{1}{\beta^3 \sigma^2 \epsilon} \cdot n^2 \cdot \log^4(n/\beta) \right) \right)$ *time.*

*Proof.* We walk through each step in Algorithm 1. Recall that each semi-distance query $\hat{w}_i(H_j)$ takes $\Theta(s)$ to compute.

First, we draw $s$ samples from $P$. As we go through the algorithm, we estimate the semi-distances $\hat{w}_i(H_j)$ as needed using these $s$ samples. We begin by setting each $\hat{W}_A(H_i)$ to be 0. The process of drawing samples and initializing proxy estimates takes $\Theta(s + n)$ time.

The algorithm runs in at most $T$ rounds. During a single round, it performs the following actions:

1. Creates $Q$ and draws $k$ samples from the exponential mechanism. Computing the probability of every $H_i$ according to $Q$ takes $\Theta(n \cdot s)$ time. Obtaining $k$ samples from $Q$ can be done in $\Theta(n + k \cdot \log k)$ time via computing the CDF and inverse sampling.

2. Invokes FIND-PROMPTING-HYPOTHESIS (Algorithm 3). For each hypothesis $H_j \in \mathcal{H}$, we compute its $\mathsf{score}_{\eta, \mathcal{K}, D}(H_j)$. Computing the score involves computing the lift value of every hypothesis in $\mathcal{K}$ and then sorting, which takes $\Theta(k \cdot s + k \cdot \log k)$ time. Notice that our choice of $s$ in Equation 31 dominates over our choice of $\log k$. Therefore, FIND-PROMPTING-HYPOTHESIS takes $\Theta(k \cdot n \cdot s)$ time.

3. Updates all $n$ proxy distances (unless this is the last round). This takes $\Theta(n \cdot s)$ time.

Therefore, each round takes $\Theta(k \cdot n \cdot s)$.

Finally, when the algorithm halts, it samples a distribution from $Q$ as output. Given there are at most $T$ iterations, the total time spent is $\Theta(T \cdot k \cdot n \cdot s)$. Substituting the values of $s, T$, and $k$ (Equations 31, 32, 33) yields the desired result. □

# E Computing the Prompting Scores

The wrapper algorithm iteratively seeks a hypothesis $H_i \in \mathcal{H}$ that can lift a significant portion of other hypotheses in $\mathcal{H}$. Concretely, we want to characterize the following cases, when $H_j$ is drawn from $Q$:

$$\mathbf{Pr}_{H_j \sim Q}\left[\hat{w}_i(H_j) - \hat{W}_A(H_j) \geq \sigma'\right] \geq \eta \quad \text{vs.} \quad \mathbf{Pr}_{H_j \sim Q}\left[\hat{w}_i(H_j) - \hat{W}_A(H_j) \geq \sigma'\right] < \frac{\eta}{4}. \quad (36)$$

Computing the exact probabilities in Equation 36 is costly, so we estimate them by sampling. Specifically, we draw a list of hypotheses $\mathcal{K} = [H_{j_1}, \cdots, H_{j_k}]$ from $Q$, where $k$ is the number of samples. We compute the empirical fraction of hypotheses in $\mathcal{K}$ that $H_i$ lifts significantly. If $k$ is sufficiently large, then the empirical estimate closely approximates the probability.

However, if we naively count the number of hypotheses in $\mathcal{K}$ who have lift values above a threshold $\sigma'$, then the result is highly sensitive: a single change in the dataset could shift every lift value from below $\sigma'$ to above $\sigma'$, thus shifting the count from 0 to $n$. To reduce sensitivity, we instead calculate an empirical quantile of the lift values. Specifically, we use the $\eta/2$-quantile of all lift values that $H_i$ induces for sampled hypotheses in $\mathcal{K}$. We call this value $\mathsf{score}_{\eta, \mathcal{K}, D}(H_i)$, which is an approximation of the probabilities in Equation 36 and it allows us to distinguish between the two cases. In particular, if $\mathsf{score}_{\eta, \mathcal{K}, D}(H_i)$ is $\eta/4$-close to the true probability, then we guarantee:

- If $H_i$ can $\sigma'$-lift $H_j$ with probability at least $\eta$, then $\mathsf{score}_{\eta, \mathcal{K}, D}(H_i)$ is at least $\sigma'$.
- If $H_i$ can $\sigma'/2$ -lift $H_j$ with probability less than $\eta/4$, then $\mathsf{score}_{\eta, \mathcal{K}, D}(H_i)$ is less than $\sigma'/2$.

In Section E.1, we prove in Lemma E.4 that the output of Algorithm 2 has low sensitivity with respect to changes in the input dataset $D$. In Section E.2, we prove that if the number of hypotheses sampled is large enough, then the value of $\mathsf{score}_{\eta, \mathcal{K}, D}(H_i)$ accurately approximates how often $H_i$ could lift hypotheses in $\mathcal{H}$ by $\sigma'$.

---

**Algorithm 2** Compute $\mathsf{score}_{\eta, \mathcal{K}, D}(H_i)$

---

1: **procedure** COMPUTE-SCORE($H_i, \eta, \mathcal{K}, D$)
2:     $\mathcal{T} = []$         ▷ initialize a list to store lift values induced by $H_i$ for each $H_{j_\ell} \in \mathcal{K}$
3:     **for** $H_{j_\ell} \in \mathcal{K}$ **do**
4:         Append $\hat{w}_i(H_{j_\ell}) - \hat{W}(H_{j_\ell})$ to $\mathcal{T}$ ▷ assume query access to $\hat{w}_i(H_{j_\ell})$, access to $\hat{W}(H_{j_\ell})$
5:     Sort $\mathcal{T}$ in non-increasing order
6:     **return** $\mathcal{T}[\lceil \eta/2 \cdot |\mathcal{K}| \rceil]$         ▷ return $\lceil \eta/2 \cdot |\mathcal{K}| \rceil$-th largest lift value

---

## E.1 Sensitivity of the score

In the following lemmas, we compute sensitivities to support the privacy analysis of our scoring mechanism. Throughout, we consider sensitivity with respect to the private dataset $D$; all other inputs to each function are assumed to be public and fixed. Lemma E.1 shows that any quantile of a sorted list can change by at most the size of the individual perturbations of elements in that list. Lemma E.2 shows the sensitivity of the empirical semi-distance $\hat{w}_i(H_j)$ by $1/s$, and Lemma E.3 shows that the sensitivity of $\hat{W}_A(H_j)$ is also $1/s$. Finally, Lemma E.4 combines these results to prove that the overall score function $\mathsf{score}_{\eta, \mathcal{K}, D}(H_i)$ has sensitivity $2/s$.

**Lemma E.1.** *Let $x = [x_1, \ldots, x_n]$ be a sorted non-increasing list. For all $i \in [n]$, let $x_i' := x_i + \delta_i$, where $|\delta_i| \leq \Delta$. Sort the set $\{x_i'\}_{i=1}^n$ into a non-increasing list $y = [y_1, ..., y_n]$. Then, for all $i \in [n]$, $|y_i - x_i| \leq \Delta$.*

*Proof.* Fix $i \in [n]$. We claim $y_i \leq x_i + \Delta$. Suppose for a contradiction that $y_i > x_i + \Delta$. Define the set of indices

$$S := \{j \in [n] : x'_j > x_i + \Delta\}.$$

Because $y$ is ordered, there are $i$ values in $y$, namely $y_1, \ldots, y_i$, that must be greater than $x_i + \Delta$. Therefore, there must be at least $i$ values of $j$ such that $x'_j > x_i + \Delta$, so $|S| \geq i$. However, notice that if $k \geq i$, then $x'_k \notin S$ because

$$x'_k \leq x_k + \Delta \leq x_i + \Delta.$$

Therefore, only indices $j < i$ can be contained in $S$, which is a contradiction since there are only $i - 1$ such indices. The other direction $y_i \geq x_i - \Delta$ is proved similarly. To verify that this bound is tight, consider $\delta_i = \Delta$ for all $i \in [n]$. $\square$

**Lemma E.2.** *Let $H_i, H_j$ be two hypotheses in $\mathcal{H}$. With respect to the dataset $D$, the sensitivity of $\hat{w}_i(H_j)$ is $1/s$.*

*Proof.* Notice that $H_j(\mathcal{S}_{i,j})$ has no dependence on $D$. However, $\hat{P}(\mathcal{S}_{i,j})$ can vary by at most $1/s$ depending on if the differing data point is in $\mathcal{S}_{i,j}$. Therefore, $\Delta(\hat{w}_i(H_j)) = 1/s$. $\square$

**Lemma E.3.** *Let $H_j \in \mathcal{H}$ and $A \subseteq \mathcal{H}$. With respect to the dataset $D$, the sensitivity of $\hat{W}_A(H_j) = \max_{H_k \in A} \hat{w}_k(H_i)$ is $1/s$.*

*Proof.* Notice that $\hat{W}_A(H_j)$ is a maximum taken over a set of empirical semi-distances. Since the maximum is a particular quantile, by Lemma E.1 and Lemma E.2, the sensitivity of $\hat{W}_A(H_j)$ is $1/s$. $\square$

**Lemma E.4.** *Fix $H_i \in \mathcal{H}$. Let $\mathcal{K}$ be a public list that consists of hypotheses in $\mathcal{H}$. In other words, consider $\mathcal{K}$ to be given and fixed. With respect to the dataset $D$, the sensitivity of $\textbf{score}_{\eta,\mathcal{K},D}(H_i)$ is $2/s$. Precisely,*

$$\Delta(\textbf{score}_{\eta,\mathcal{K},D}(H_i)) = \sup_{\substack{D,D' \in \mathcal{X}^{\otimes s} \\ \textbf{Ham}(D,D')=1}} |\textbf{score}_{\eta,\mathcal{K},D}(H_i) - \textbf{score}_{\eta,\mathcal{K},D'}(H_i)| = 2/s.$$

*Proof.* Consider the sensitivity of the lift value $H_i$ induces on $H_j$, defined as $\hat{w}_i(H_j) - \hat{W}_A(H_j)$. By Lemma E.2 and Lemma E.3, the sensitivity of each term is $1/s$, so the sensitivity of $\hat{w}_i(H_j) - \hat{W}_A(H_j)$ is $2/s$. Since Algorithm 2 returns a fixed quantile of the lift values, and each lift value has sensitivity at most $2/s$, Lemma E.1 implies that $\textbf{score}_{\eta,\mathcal{K},D}(H_i)$ also has sensitivity $2/s$. $\square$

## E.2 Accuracy of the score

In the following lemma, we discuss the accuracy of the score. In fact, we show that the score helps us to distinguish the two cases defined in Equation 36.

**Lemma E.5.** *Let $\mathcal{K} = [H_{j_1}, ..., H_{j_k}]$ be a list of hypotheses, where each $H_{j_\ell} \in \mathcal{K}$ represents an i.i.d. sample from $Q$. If $k$ is at least*

$$\frac{12 \log(n/\beta_{SCO})}{\eta},$$

*then with probability at least $1 - \beta_{SCO}$ (taken over the randomness of $H_{j_\ell}$'s) the following holds for every $H_i \in \mathcal{H}$:*

1. *If $H_i$ is $(\sigma', \eta)$-prompting with respect to $Q$, then $\textbf{score}_{\eta,\mathcal{K},D}(H_i)$ is at least $\sigma'$.*

2. *If $H_i$ is not $(\sigma'/2, \eta/4)$-prompting with respect to $Q$, then $\textbf{score}_{\eta,\mathcal{K},D}(H_i)$ is less than $\sigma'/2$.*

*Proof.* Fix a candidate hypothesis $H_i \in \mathcal{H}$. For each $H_{j_\ell} \in \mathcal{K}$, use an indicator variable $\mathbb{1}_{\hat{w}_i(H_{j_\ell}) - \hat{W}_A(H_{j_\ell}) \geq t}$, where $t \in [0, 1)$, to determine whether $H_i$ can lift $H_j$ by at least $t$ or not.

Notice the expectation of this indicator variable evaluates to the probability that $H_i$ lifts $H_j$ by at least $t$,

$$\mathbf{E}_{H_{j_\ell} \sim Q}\left[ \mathbb{1}_{\hat{w}_i(H_{j_\ell}) - \hat{W}_A(H_{j_\ell}) \geq t} \right] = \mathbf{Pr}_{H_j \sim Q}\left[ \hat{w}_i(H_j) - \hat{W}_A(H_j) \geq t \right].$$

We set $t = \sigma'$ in case 1 and $\sigma'/2$ in case 2. We estimate the probability above using the following empirical estimators, $\bar{Z}_i^{\sigma'}$ and $\bar{Z}_i^{\sigma'/2}$, defined as:

$$\bar{Z}_i^{\sigma'} := \frac{1}{k} \sum_{\ell=1}^{k} \mathbb{1}_{\hat{w}_i(H_{j_\ell}) - \hat{W}_A(H_{j_\ell}) \geq \sigma'}$$

and

$$\bar{Z}_i^{\sigma'/2} := \frac{1}{k} \sum_{\ell=1}^{k} \mathbb{1}_{\hat{w}_i(H_{j_\ell}) - \hat{W}_A(H_{j_\ell}) \geq \sigma'/2}$$

1. If $H_i$ is $(\sigma', \eta)$-prompting with respect to $Q$, then

$$\mathbf{Pr}_{H_j \sim Q}\left[ \hat{w}_i(H_j) - \hat{W}_A(H_j) \geq \sigma' \right] \geq \eta.$$

Using a Chernoff bound, we obtain

$$\mathbf{Pr}_{H_{j_\ell} \sim Q}\left[ \bar{Z}_i^{\sigma'} \leq \frac{\eta}{2} \right] \leq \mathbf{Pr}_{H_{j_\ell} \sim Q}\left[ \bar{Z}_i^{\sigma'} \leq \left(1 - \frac{1}{2}\right) \mathbf{E}_{H_{j_\ell} \sim Q}\left[ \mathbb{1}_{\hat{w}_i(H_{j_\ell}) - \hat{W}_A(H_{j_\ell}) \geq \sigma'} \right] \right]$$

$$\leq \exp\left( -k \, \mathbf{E}_{H_{j_\ell} \sim Q}\left[ \mathbb{1}_{\hat{w}_i(H_{j_\ell}) - \hat{W}_A(H_{j_\ell}) \geq \sigma'} \right]/8 \right)$$

$$\leq \exp\left( -k\eta/8 \right) \leq \frac{\beta_{SCO}}{n}.$$

Therefore, with probability $\geq 1 - \beta_{SCO}/n$, at least $\eta/2$ fraction of the hypotheses in $\mathcal{K}$ can be lifted by $H_i$ by at least $\sigma'$. This implies $H_i$ is $(\sigma', \eta/2)$-empirical-prompting with respect to $\mathcal{K}$. This further implies that there are at least $\lceil k\eta/2 \rceil$ entries among the lift values in $\mathcal{T}$ that are at least $\sigma'$. Therefore, the $\lceil k\eta/2 \rceil$-th largest lift values must be at least $\sigma'$. Thus, $\mathsf{score}_{\eta,\mathcal{K},D}(H_i)$ is at least $\sigma'$ as desired in the statement of the lemma.

2. If $H_i$ is not $(\sigma'/2, \eta/4)$-prompting with respect to $Q$, then

$$\mathbf{Pr}_{H_j \sim Q}\left[ \hat{w}_i(H_j) - \hat{W}_A(H_j) \geq \sigma'/2 \right] < \eta/4.$$

Take $X$ as a binomial random variable with parameter $(k, \eta/4)$. Clearly, $X/k$ is stochastically larger than $\bar{Z}_i^{\sigma'/2}$, meaning for any fix threshold $x \in (0, 1]$, the probability of $X/k > x$ is larger than the probability of $\bar{Z}_i^{\sigma'/2} > x$. Setting $x = \sigma'/2$ attains

$$\mathbf{Pr}_{H_{j_\ell} \sim Q}\left[ \bar{Z}_i^{\sigma'/2} > \eta/2 \right] < \mathbf{Pr}_{X \sim \mathbf{Bin}(k,\eta/4)}\left[ \frac{X}{k} > \eta/2 \right]$$

$$= \mathbf{Pr}_{X \sim \mathbf{Bin}(k,\eta/4)}\left[ \frac{X}{k} > (1+1) \cdot \mathbf{E}\left[ \frac{X}{k} \right] \right]$$

$$\leq \exp\left( -\frac{k\eta}{12} \right) \leq \frac{\beta_{SCO}}{n}.$$

Thus, with probability $\geq 1 - \beta_{SCO}/n$, fewer than $\eta/2$ fraction of lift values exceed $\sigma'$. This implies that $H_i$ is not $(\sigma', \eta/2)$-empirical-prompting with respect to $\mathcal{K}$. This further implies that there are less than $\lceil k\eta/2 \rceil$ entries of lift values at least $\sigma'$. Thus, $\mathsf{score}_{\eta,\mathcal{K},D}(H_i)$ is less than $\sigma'$.

Using a union bound, we can show the above holds for all the hypotheses in $\mathcal{H}$. Hence, the proof is complete.

$\square$

## F  Finding a Prompting Hypothesis

To privately find a prompting hypothesis given the scores computed in Algorithm 2, we use the sparse vector technique [DR+14, LSL16]. We feed a stream of scores into Algorithm 3, which privately outputs either the index of the hypothesis which was detected to have a score above $\frac{3\sigma'}{4}$, or $\perp$, if no hypotheses have sufficiently high scores. With high probability, we can guarantee that, if the mechanism outputs $i$, then $\mathsf{score}_{\eta,\mathcal{K},D}(H_i) > \frac{\sigma'}{2}$, and, if the mechanism sees a hypothesis $H_i$ with $\mathsf{score}_{\eta,\mathcal{K},D}(H_i) > \sigma'$, it will not output $\perp$.

---

**Algorithm 3** An algorithm for privately finding a prompting hypothesis

1: **procedure** FIND-PROMPTING-HYPOTHESIS($\epsilon, \Delta, \sigma', \eta, \mathcal{H}, \mathcal{K}, D$)
2:     $\epsilon_1 \leftarrow \frac{\epsilon}{2}$
3:     $\epsilon_2 \leftarrow \epsilon - \epsilon_1$
4:     $\rho \leftarrow \mathbf{Lap}\left(\frac{\Delta}{\epsilon_1}\right)$
5:     $\tau \leftarrow \frac{3\sigma'}{4}$
6:     $\hat{\tau} \leftarrow \tau + \rho$
7:     **for** $H_i \in \mathcal{H}$ **do**
8:         $\nu_i \leftarrow \mathbf{Lap}\left(\frac{2\Delta}{\epsilon_2}\right)$
9:         $\mathsf{score}_{\eta,\mathcal{K},D}(H_i) \leftarrow \text{COMPUTE-SCORE}(H_i, \eta, \mathcal{K}, D)$       ▷ Algorithm 2
10:         **if** $\mathsf{score}_{\eta,\mathcal{K},D}(H_i) + \nu_i \geq \hat{\tau}$ **then**
11:             **return** $H_i$ and **halt**
12:     **return** $\perp$ and **halt**

---

**Theorem 6** (Theorems 3.23 and 3.24 of [DR+14]). *Suppose we are given parameters $\epsilon, \Delta > 0$ and $\sigma', \eta \in (0, 1]$. Assume we are given two lists of hypotheses $\mathcal{H}$ and $\mathcal{K}$ such that $\mathsf{score}_{\eta,\mathcal{K},D}(\cdot)$ has sensitivity at most $\Delta \leq \frac{\sigma' \epsilon_2}{32 \log(2/\beta_{SVT})}$. Then the FIND-PROMPTING-HYPOTHESIS Procedure in Algorithm 3 receives $\epsilon, \Delta, \sigma', \eta, \mathcal{H}, \mathcal{K},$ and $D$ as its input and outputs $H_i$ or $\perp$ with $\epsilon$-privacy such that, with probability at least $1 - \beta_{SVT}$:*

1. *If* FIND-PROMPTING-HYPOTHESIS *outputs $H_i \in \mathcal{H}$, then $\mathsf{score}_{\eta,\mathcal{K},D}(H_i) > \frac{\sigma'}{2}$.*

2. *If there exists $H_i$ such that $\mathsf{score}_{\eta,\mathcal{K},D}(H_i) > \sigma'$, then* FIND-PROMPTING-HYPOTHESIS *does not output $\perp$.*

**Proof overview:** Theorem 6 follows from Theorems 3.23 and 3.24 in [DR+14]. The privacy guarantee and the first statement of the theorem's accuracy guarantee are stated directly in [DR+14], while the second accuracy statement is the contrapositive of the second statement in [DR+14], Theorem 3.24.

*Proof.* FIND-PROMPTING-HYPOTHESIS is an instance of the sparse vector technique [DR+14]. When $\Delta$ is an upper bound on the sensitivity of the outputs of COMPUTE-SCORE, the privacy of the algorithm holds by Theorem 3.23 in [DR+14].

The proof of accuracy of FIND-PROMPTING-HYPOTHESIS is a simple modification of Theorem 3.24 in [DR+14], removing the requirement that the last query (or, in this case, hypothesis) is the only query with a score close to being above the threshold.

Let $\tau = \frac{3\sigma'}{4}$ be the threshold of the mechanism. We want to find conditions on $\sigma'$ such that if Algorithm 3 outputs $H_i$, then $\mathsf{score}_{\eta,\mathcal{K},D}(H_i) > \tau - \frac{\sigma'}{4} = \frac{\sigma'}{2}$, and, if Algorithm 3 outputs $\perp$, then for all $i$, $\mathsf{score}_{\eta,\mathcal{K},D}(H_i) < \tau + \frac{\sigma'}{4} = \sigma'$. We will then use these conditions to establish bounds on $\Delta$. Note that the second statement here is the contrapositive of the second statement in our theorem statement.

Observe that it is sufficient to find conditions on $\sigma'$ such that, with probability at most $1 - \beta_{SVT}$:

$$\max_{i \in [n]} |\nu_i| + |\rho| \leq \frac{\sigma'}{4}. \tag{37}$$

Recall that we don't halt at $i$ if:

$$
\begin{aligned}
&\mathsf{score}_{\eta,\mathcal{K},D}(H_i) + \nu_i < \tau + \rho \\
\implies &\mathsf{score}_{\eta,\mathcal{K},D}(H_i) < \tau + \rho - \nu_i \\
&\qquad\qquad\qquad\quad \leq \tau + |\rho| + |\nu_i| \\
&\qquad\qquad\qquad\quad \leq \tau + \frac{\sigma'}{4} \qquad \text{by Equation 37} .
\end{aligned}
$$

Further, if we do halt at $i$, then:

$$
\begin{aligned}
&\mathsf{score}_{\eta,\mathcal{K},D}(H_i) + \nu_i \geq \tau + \rho \\
\implies &\mathsf{score}_{\eta,\mathcal{K},D}(H_i) \geq \tau + \rho - \nu_i \\
&\qquad\qquad\qquad\quad \geq \tau - (|\rho| + |\nu_i|) \\
&\qquad\qquad\qquad\quad \geq \tau - \frac{\sigma'}{4} \qquad \text{by Equation 37} .
\end{aligned}
$$

Thus, to find $\sigma'$ satisfying Equation 37, we can equivalently find conditions on $\sigma'$ such that:

$$
\mathbf{Pr}\left[|\rho| \geq \frac{\sigma'}{8}\right] \leq \frac{\beta}{2} \qquad \text{and} \qquad \mathbf{Pr}\left[\max_{i\in[n]} |\nu_i| \geq \frac{\sigma'}{8}\right] \leq \frac{\beta}{2} .
$$

By the properties of the Laplace distribution and the union bound, we find $\frac{\sigma'}{4} \geq \frac{8\Delta}{\epsilon_2} \log\left(\frac{2n}{\beta_{\mathrm{SVT}}}\right)$.

This results in a bound on $\Delta$ of:

$$
\Delta \leq \frac{\sigma'\epsilon}{32 \log\left(2n/\beta_{\mathrm{SVT}}\right)} .
$$

$\qquad\qquad\qquad\qquad\qquad\qquad\qquad\qquad\qquad\qquad\qquad\qquad\qquad\qquad\qquad\qquad\quad \square$

