# OpenReview forum: "Nearly-Linear Time Private Hypothesis Selection with the Optimal Approximation Factor"
_NeurIPS.cc/2025/Conference — NeurIPS 2025 poster_

### Official Review · Reviewer_5ALS · 2025-06-16

**Clarity:** 4
**Significance:** 3
**Originality:** 3
**Rating:** 5
**Confidence:** 4

**Summary:**

This work investigates the hypothesis selection problem under differential privacy. Given a list of $n$ hypotheses (candidate distributions), the objective is to select one that approximates the unknown underlying distribution while preserving data privacy. In this paper, the authors propose an algorithm that achieves an optimal (even without privacy) approximation factor $\alpha = 3$ with nearly-linear runtime $\tilde{O}(n)$. This improves existing algorithms in prior work, which typically run in at least quadratic time. Notably, the sample complexity of the proposed method scales poly-logarithmically in $n$, which is only a mild increase compared to the non-private results.

**Questions:**

1. The time complexity of the proposed algorithm includes a $1/\sigma$ factor while previous methods do not. Is it possible to eliminate this dependence via allowing a larger approximation factor $\alpha$ like in [ABS24]?
2. I find the notations $W(H_j)$ and $\tilde{W}(H_j)$ somewhat confusing. They seem to be defined w.r.t. the actual distribution $P$, whereas in many places they should represent the distance w.r.t. the empirical distribution $\hat{P}$ (e.g., in Algorithm 1 and its analysis, particularly in Theorem 4). Could you clarify their precise definitions?
3. The authors apply the SVT in Algorithm 3 to privately identify a prompting hypothesis with a high score. I wonder why not directly invoke a report-noisy-max mechanism, which yields a much simpler analysis.
4. In Lemma F.3, should $w_k(H_i)$ be $\hat{w}_k(H_i)$?
5. Should the $\sigma'$ in the second statement in Lemma F.5 be replaced by $\sigma' / 2$ (for consistency with Line 640 and 641)? Additionally, the $H_j$ in the proof should be $H_{j_\ell}$ and $t$ should be $k = \lvert\mathcal{K}\rvert$.

**Ethical Concerns:**

["NO or VERY MINOR ethics concerns only"]

**Final Justification:**

While I'm uncertain about the significance of the main technical contribution, the paper does propose an algorithm for the private hypothesis selection problem with an optimal approximation ratio. Also, the utility bounds match the state-of-the-art non-private algorithm.

**Limitations:**

yes

**Quality:**

3

**Strengths And Weaknesses:**

Strengths:

1. This work introduces a nearly-linear time algorithm for private hypothesis selection, significantly improving the runtime compared to prior methods.
2. The proposed algorithm achieves an approximation ratio of $\alpha = 3$, which is known to be optimal even without privacy. Moreover, it results in a sample complexity of only $\mathrm{polylog}(n)$.
3. The article is well-organized and the writing is clear and concise.

Weakness:

1. While the algorithm incurs a sample complexity of only $\mathrm{polylog}(n)$, non-private algorithms require only $O(\log n)$ samples, indicating a notable drawback.
2. The dependence on the confidence $\beta$ and additive error $\sigma$ in the sample complexity and time complexity bounds is greater than other algorithms (with worse $\alpha$ or runtime). This may become a serious issue if one aims for a high success probability or a small additive error.
3. It seems the algorithm is a simple extension of [ABS24]'s with standard techniques from differential privacy. The only thing I find interesting is analyzing the evolution of the exponential mechanism (Lemma D.2).

---

> ### Author Rebuttal · Authors · 2025-07-30
>
> We thank the reviewer for their detailed feedback and constructive comments. We address the concerns and questions below.
>
> ---
>
> ### **Polylogarithmic Sample Complexity (Weakness 1)**
>
> > While the algorithm incurs a sample complexity of only $\operatorname{polylog}(n)$, non-private algorithms require only samples $\log(n)$, indicating a notable drawback.
>
> We agree that the gap between our $\operatorname{polylog}(n)$ sample complexity and the non-private $\log(n)$ is an important point.
>
> Our primary contribution is introducing a new stable, private minimum distance estimator that avoids direct comparisons (as in prior tournament-style algorithms) while obtaining a near-linear time private algorithm. To obtain the desired time complexity, our algorithm must *not* make very many wasteful queries, and it must adaptively choose the next hypothesis to query depending on previous outcomes.
>
> Unfortunately, this adaptivity is challenging to privatize because, in principle, the decision of which hypothesis to choose makes the algorithm very sensitive. Thus, we have to make every step private. The additional $\log^2(n)$ factors are a natural byproduct of reconciling the inherent adaptivity of the algorithm with the need for privacy.
>
> Whether these $\operatorname{polylog}$ factors are a necessary price to pay to achieve near-linear time complexity is an interesting open direction. We hope our result paves the way for more progress on this topic.
>
> ### **Dependence on** $\beta$ **and** $\sigma$ **(Weakness 2)**
>
> > The dependence on the confidence and additive error in the sample complexity and time complexity bounds is greater than other algorithms (with worse or runtime). This may become a serious issue if one aims for a high success probability or a small additive error.
>
> We acknowledge that an ideal algorithm should have a sample complexity and time complexity with respect to $\beta$ of $O(\log(1/β))$, which our current algorithm does not achieve. Specifically, our algorithm has an additional $1/\beta^2$ in the sample complexity as well as a $1/\beta^4$ in the time complexity.
>
> Even in the non-private case, this $\beta$ dependence is difficult to control with algorithms that obtain $\alpha < 9$, largely because there is no known "amplification process" for hypothesis selection that preserves the approximation factor $\alpha$, as opposed to many learning theory problems. See Lines 123-130 for more details.
>
> For example, the first algorithm of [ABS24] ($\alpha = 3$) obtains a time complexity of $1/\beta^3$. The random ladder tournament in [ABS23] incurs a factor of $1/\beta$ in the sample complexity while obtaining $\alpha = 5$. Despite this dependence on $\beta$, we believe our result is an important step forward as the first nearly-linear time algorithm for private hypothesis selection that obtains the optimal $\alpha$.
>
> It is important to note that if we relax the requirement for $\alpha$ and allowing for a worse approximation factor, this dependence on $\beta$ may be improved. The second algorithm of [ABS24] ($\alpha = 4$) obtains a time complexity without polynomial dependence on $1/\beta$, and it is the only near-linear time algorithm with $\alpha < 9$. We did not focus on this algorithm due to the sub-optimality of its approximation factor. It remains an open question whether privatizing this algorithm would lead to a better dependence on $\beta$ (and $\sigma$, as the reviewer suggests in question 1).
>
> With respect to $\sigma$, our bound matches those of the $\alpha=3$ non-private algorithm in [ABS24]. While improving these dependencies is a valuable direction for future work (even in the non-private case), our result does not exacerbate these known challenges.
>
> ### **Major Differences Compared to [ABS24] (Weakness 3)**
> > It seems the algorithm is a simple extension of [ABS24]'s with standard techniques from differential privacy. The only thing I find interesting is analyzing the evolution of the exponential mechanism (Lemma D.2).
>
> We appreciate the reviewer's observation and acknowledge that the structure of our algorithm is indeed closely aligned to that of [ABS24]. However, we note that other nearly-linear algorithms for hypothesis selection (e.g., [AJOS14, AFJ+18]) are tournament-based and require many pairwise comparisons. Such comparisons are highly sensitive and therefore difficult to privatize. We chose to design a private algorithm based on the structure of [ABS24] to avoid these direct comparisons.
>
> Nevertheless, despite not following a tournament-based structure, many components of [ABS24]’s algorithm are highly sensitive—particularly its bucketing scheme. In our private algorithm, we replace this bucketing scheme with an exponential mechanism, $Q$, that skews towards sampling hypotheses with low proxy distances. Instead of choosing a prompting hypothesis based on its ability to lift many hypotheses in a bucket, we choose a prompting hypothesis based on its ability to lift hypotheses with a large probability mass under the exponential mechanism. As a result, we can only guarantee that, for each selected prompting hypothesis, only the proxy distances of a few (or even one) hypothesis will increase. Both our algorithm and [ABS24] require $O(n)$ time to identify each prompting hypothesis. While [ABS24] prove that their algorithm terminates in near-linear time by showing that only $O(\log n)$ prompting hypotheses can be identified at each bucket, we need a clever way of monitoring the progress of the algorithm to ensure that our algorithm similarly ends after finding $O(\log n)$ prompting hypotheses. This is the most significant challenge that we overcome, in Section D.2.
>
> ### **Factor of $1/\sigma$ (Question 1)**
>
> > The time complexity of the proposed algorithm includes a $1/\sigma$ factor while previous methods do not. Is it possible to eliminate this dependence via allowing a larger approximation factor like in [ABS24]?
>
> This is a great question for future work. We believe it is feasible to privatize the second algorithm in [ABS24] with $\alpha = 4$. However, we did not focus on this algorithm because of its sub-optimal approximation factor.
>
> ### **Notation (Question 2)**
>
> > I find the notations $W(H_j)$ and $\tilde{W}(H_j)$ somewhat confusing. They seem to be defined w.r.t. the actual distribution $P$, whereas in many places they should represent the distance w.r.t. the empirical distribution (e.g., in Algorithm 1 and its analysis, particularly in Theorem 4). Could you clarify their precise definitions?
>
> Thank you for pointing this out. Both the semi-distance $w_i(H_j)$ and the maximum semi-distance $W(H_j) := \max_{H_i \in \mathcal{H}}w_i(H_j)$ are defined with respect to the actual distribution $P$. On Line 189, $\tilde{W}(H_j)$ is erroneously defined as $\max_{H_i \in A} w_i(H_j)$. This will be corrected to $\max_{H_i \in A} \hat{w}_i(H_j)$ in the revision. Throughout the rest of our paper, each $\hat{w}_i(H_j)$ is an "empirical approximation" of ${w}_i(H_j)$ and is computed empirically from samples of $P$, as in Algorithm 1 and Theorem 4.
>
> ### **Using Report-Noisy-Max (Question 3)**
> > The authors apply the SVT in Algorithm 3 to privately identify a prompting hypothesis with a high score. I wonder why not directly invoke a report-noisy-max mechanism, which yields a much simpler analysis
>
> Thank you for your suggestion. In our analysis, we need each call to Find-Prompting-Hypothesis to satisfy two criteria in order for the algorithm to succeed: first, if we return a hypothesis, that hypothesis must have a sufficiently high score; and second, if we don't return any hypothesis, then all hypotheses must have sufficiently low scores. We apply SVT as a off-the-shelf procedure that gives these guarantees, as proven with only a very simple modification to existing results [DR+14]. If report-noisy-max also satisfies Theorem 6—which we believe it would, provided that we take care of the case where the maximum score is below our threshold—you are correct to assert that it could take the place of SVT in our algorithm. With this additional case, this would also be a simple modification of the existing result in the literature.
>
> ### **Typos (Questions 4 and 5)**
>
> > In Lemma F.3, should $w_k(H_i)$ be $\hat{w}_k(H_i)$?
>
> > Should the $\sigma'$ in the second statement in Lemma F.5 be replaced by $\frac{\sigma'}{2}$ (for consistency with Line 640 and 641)? Additionally, the $H_j$ in the proof should be $H_{j_{l}}$ and $t$ should be $k = |K|$
>
> Thank you for pointing out these typos—we appreciate your close attention to details. We will revise the notation in Lemmas F.3 and F.5 accordingly.
>
> ---
>
> ### **References**
>
> [ABS23] Maryam Aliakbarpour, Mark Bun, and Adam Smith. Hypothesis selection with memory constraints, NeurIPS, 2023.
>
> [ABS24] Maryam Aliakbarpour, Mark Bun, and Adam Smith. Optimal hypothesis selection in (almost) linear time. NeurIPS, 2024.
>
> [AFJ+18] Jayadev Acharya, Moein Falahatgar, Ashkan Jafarpour, Alon Orlitsky, and Ananda Theertha Suresh. Maximum selection and sorting with adversarial comparators. The Journal of Machine Learning Research, 2018.
>
> [AJOS14] Jayadev Acharya, Ashkan Jafarpour, Alon Orlitsky, and Ananda Theertha Suresh. Sorting with adversarial comparators and application to density estimation. IEEE International Symposium on Information Theory, 2014.
>
> [DR+14] Cynthia Dwork, and Aaron Roth. The algorithmic foundations of differential privacy. Foundations and Trends in Theoretical Computer Science, 2014.

---

> ### Comment · Reviewer_5ALS · 2025-08-05
>
> Thank you for your response. While I'm uncertain about the significance of the main technical contribution (replacing the bucketing scheme with an exponential mechanism and proving the algorithm terminates in $O(\log n)$ rounds), I appreciate your clarification on the other questions.
>
> I will raise my score to 5.

---

### Official Review · Reviewer_TVoF · 2025-06-25

**Clarity:** 3
**Significance:** 3
**Originality:** 3
**Rating:** 5
**Confidence:** 2

**Summary:**

This paper studies the problem of hypothesis selection under differential privacy: Given a set of "hypotheses" (distributions) $\\mathcal{H} = \\{H\_1, H\_2, \\ldots, H\_n\\}$, and sample access to a distribution $P$, the goal is to return a hypothesis $H \\in \\mathcal{H}$ that minimizes the total variation distance to $P$, namely $\\|H - P\\|\_{\\mathrm{TV}}$. Moreover it is required that the algorithm satisfies $\\varepsilon$-differential privacy with respect to the samples from $P$.

There are three parameters of interest:
* Approximation ratio $\\alpha$: The return $H$ should satisfy $\\|H - P\\|\_{\\mathrm{TV}} \\le \\alpha \cdot \\|H^* - P\\|\_{\\mathrm{TV}} + \\sigma$, where $\\sigma$ is an additive error. Even without DP requirement, it is known that $\\alpha = 3$ is the best possible approximation ratio.
* Running time of the algorithm: Recent work has obtained a linear time (in $n$, the number of hypotheses).
* Sample complexity of the algorithm.

Previously, $O(\\log n)$ sample complexity has been obtained with DP constraint and for approximation ratio $\\alpha = 3$ (which is the best possible). However, the running time of that algorithm is quadratic in $n$.

The main result of the paper is to provide an algorithm that achieves nearly linear running time in $n$, approximation ratio $\\alpha = 3$, although with slightly larger $O(\\log^3 n)$ sample complexity.

**Questions:**

### Comments

* (Page 4: Table 1) It wasn't clear until much later what $s$ means in Table 1. (I understood later that it is the number of samples, which is basically the same as the last column. This could perhaps be made clear by having the last column header as "_Sample Complexity ($s$)_".
* (Page 5): It might be better to write $w\_i(H\_j)$ as $w\_i(H\_j; P)$ to emphasize that it is a notion of distance from $P$. Similarly, $\\tilde{W}(H\_j)$ could be $\\tilde{W}\_A(H\_j; P)$. I found it very confusing to keep track of what depends on what... (I understand that in Algorithm 1 $\\tilde{W}(H\_j)$ is used to denote a "variable", but perhaps some other notation could be used for that?)
* (Page 5): In the definition of _lift value_, I suppose it should be required that $i \\notin A$, but this is not specified.

**Ethical Concerns:**

["NO or VERY MINOR ethics concerns only"]

**Final Justification:**

I had a good opinion of the paper to being with and was supportive of acceptance. My only concern was that there might be too many similarities with [ABS24], however after author's response, I feel there is sufficient novelty in the paper. So I will maintain my rating and support acceptance.

**Limitations:**

The paper sufficiently discusses limitations and outlines open directions for future work.

I don't have any concerns about potential negative societal impacts of this work.

**Quality:**

3

**Strengths And Weaknesses:**

### Strengths
The paper makes important progress on the problem of private hypothesis selection, providing a linear time algorithm, while also obtaining the optimal approximation ratio of $\\alpha = 3$ and polylogarithmic sample complexity.

### Weaknesses
The contributions _might_ be incremental over the work of [[ABS24](https://proceedings.neurips.cc/paper_files/paper/2024/file/ffee3090eac0aae698b2d77ac5642c2c-Paper-Conference.pdf)]. The algorithm seems to follow a similar outline with key steps replaced with privatized versions of the same, although with clever use of the exponential mechanism and the sparse vector technique (so I am not certain about my "incremental" claim).

I felt that notations are a bit cumbersome and oversimplified, making it hard to follow (see comments under "Questions" below).
This also made the paper quite dense to follow (after Page 7 onwards) and relies on a fair bit of context that is only present in the Appendix.

The proofs in the paper look correct to me (the privacy proof is immediate from basic composition of exponential mechanism and sparse vector technique). But since I found the paper to be very dense and I was familiar with the prior work, I haven't been able to follow everything in detail.

---

> ### Author Rebuttal · Authors · 2025-07-30
>
> Thank you for the comments and detailed suggestions for improving the paper's notation. We will incorporate them into the final version.
>
> ---
>
> ### **Major Differences Compared to [ABS24]**
>
> > The contributions *might* be incremental over the work of [ABS24]. The algorithm seems to follow a similar outline with key steps replaced with privatized versions of the same, although with clever use of the exponential mechanism and the sparse vector technique (so I am not certain about my "incremental" claim)
>
> We appreciate the reviewer's observation and acknowledge that the structure of our algorithm is indeed closely aligned to that of [ABS24]. However, we note that other nearly-linear algorithms for hypothesis selection (e.g., [AJOS14, AFJ+18]) are tournament-based and require many pairwise comparisons. Such comparisons are highly sensitive and therefore difficult to privatize. We chose to design a private algorithm based on the structure of [ABS24] to avoid these direct comparisons.
>
> Nevertheless, despite not following a tournament-based structure, many components of [ABS24]’s algorithm are highly sensitive—particularly its bucketing scheme. In our private algorithm, we replace this bucketing scheme with an exponential mechanism, $Q$, that skews towards sampling hypotheses with low proxy distances. Instead of choosing a prompting hypothesis based on its ability to lift many hypotheses in a bucket, we choose a prompting hypothesis based on its ability to lift hypotheses with a large probability mass under the exponential mechanism. As a result, we can only guarantee that, for each selected prompting hypothesis, only the proxy distances of a few (or even one) hypothesis will increase. Both our algorithm and [ABS24] require $O(n)$ time to identify each prompting hypothesis. While [ABS24] prove that their algorithm terminates in near-linear time by showing that only $O(\log n)$ prompting hypotheses can be identified at each bucket, we need a clever way of monitoring the progress of the algorithm to ensure that our algorithm similarly ends after finding $O(\log n)$ prompting hypotheses. This is the most significant challenge that we overcome, in Section D.2.
>
>
>
> ### **Notation**
>
> > (Page 4: Table 1) It wasn't clear until much later what $s$ means in Table 1. (I understood later that it is the number of samples, which is basically the same as the last column. This could perhaps be made clear by having the last column header as "Sample Complexity ($s$)".
>
> Thank you for pointing this out. We will fix this issue in the final draft.
>
> > (Page 5): It might be better to write $w_i(H_j)$ as $w_i(H_j; P)$ to emphasize that it is a notion of distance from $P$. Similarly, $\tilde{W}(H_j)$ could be $\tilde{W}_{A}(H_j; P)$. I found it very confusing to keep track of what depends on what... (I understand that in Algorithm 1 $\tilde{W}(H_j)$ is used to denote a "variable", but perhaps some other notation could be used for that?)
>
> Thank you for suggesting a better notation for semi-distances and proxy distances. In terms of notations: $w_i(H_j)$ refers to the value of true semi-distances and $\hat{w_i}(H_j)$ refers to the estimated semi-distances. $\tilde{W}(H_j)$ refers to proxy distances, defined as $\max_{H_i \in A}\hat{w}_i(H_j)$, where $A$ is an ever-growing set of hypotheses during execution of algorithm 1. These notations were intended to follow the same patterns in previous works like [ABS24] to maintain consistency. However, we will adopt a more clear notation in the final draft.
>
> > (Page 5): In the definition of lift value, I suppose it should be required that $i \not \in A$, but this is not specified.
>
> As you stated, for hypotheses $H_i$ and $H_j$, where $H_i$ is already in $A$, we have that $\tilde{W}(H_j) := \max_{H_i \in A}\hat{w}_i(H_j)$ is greater than or equal to $\hat{w}_i(H_j)$. Hence, the hypotheses that are already in $A$ are not going to lift any hypotheses. Therefore, we exclude these hypotheses when we are searching for a prompting hypothesis as stated in line 13 of Algorithm 1. We will consider explicitly stating this requirement on page 5 to improve clarity.
>
> ---
>
> ### **References**
>
> [ABS24] Maryam Aliakbarpour, Mark Bun, and Adam Smith. Optimal hypothesis selection in (almost) linear time. NeurIPS, 2024.
>
> [AFJ+18] Jayadev Acharya, Moein Falahatgar, Ashkan Jafarpour, Alon Orlitsky, and Ananda Theertha Suresh. Maximum selection and sorting with adversarial comparators. The Journal of Machine Learning Research, 2018.
>
> [AJOS14] Jayadev Acharya, Ashkan Jafarpour, Alon Orlitsky, and Ananda Theertha Suresh. Sorting with adversarial comparators and application to density estimation. IEEE International Symposium on Information Theory, 2014.

---

> ### Comment · Reviewer_TVoF · 2025-08-05
>
> Thank you for your response. I agree there is sufficient novelty in the paper, despite high level similarities with [ABS24].
> I will maintain my rating and support acceptance.
>
> I have no further questions.

---

### Official Review · Reviewer_MJur · 2025-07-01

**Clarity:** 4
**Significance:** 4
**Originality:** 4
**Rating:** 5
**Confidence:** 4

**Summary:**

This paper looks at the hypothesis selection problem under the constraint of differential privacy. Hypothesis selection is a very fundamental problem in statistics which is also at the core of many distribution learning algorithms. The goal is as follows. Given a finite set of M distributions and samples from an unknown distribution, return a hypothesis which is within O(1) of the best hypothesis.

Such a problem has been studied a lot both in the private and non-private setting. It was only recently that there was a fast (i.e. near-linear time) algorithm known for the non-private setting with the optimal constant. The main result of this paper is to extend this result to the private setting. However, note that in doing so, this paper actually requires increasing the sample complexity by polylogarithmic factors. Nonetheless, this makes substantial progress in a very important problem in differentially private statistics.

The algorithm and its analysis appear to be quite nice but sophisticated. The building block is the standard Scheffe-style argument. In the Scheffe argument, one essentially tries to obtain witnesses to prove that a hypothesis is not that great (i.e. a witness to prove large TV distance). My understanding of the paper is that they start with poor witnesses. Then they try to “prompt” (the term the authors use) to find a hypothesis that increases the witness of other hypotheses. Exactly determining the hypothesis would likely violate privacy so the authors have to do this approximately. For this, they make use of the sparse vector technique which will privately find such a hypothesis. They show that the algorithm terminates when SVT fails and then one can then use the exponential mechanism using scores based on the TV distances to return a good candidate.

The analysis of the running time seems nice as well. They argue that it’s not hard to argue that it terminates in O(n) rounds but this would result in poor running time. Instead, they show that it can terminate in O(log n) rounds to get O(n log n) running time.

**Questions:**

No questions

**Ethical Concerns:**

["NO or VERY MINOR ethics concerns only"]

**Final Justification:**

Overall good paper. While a lot of the paper is still based on ABS, making the algorithm private seems to still be non-trivial.

I listed some minor technical downsides but these are more for potential follow up work.

**Limitations:**

Yes

**Paper Formatting Concerns:**

No issues

**Quality:**

4

**Strengths And Weaknesses:**

**Strengths**
- Fundamental problem in differentially private statistics.
- Makes significant contribution to the literature but showing that there exists a nearly linear time algorithm for the problem with optimal constant albeit with slightly worse sample complexity than optimal.
- Paper is written very well. It was easy to grasp the main ideas of the algorithm. The authors did an excellent job bringing me up to speed with some of the literature.
- The techniques seem very nice. I would personally love to learn more about the paper and to talk to the authors at a poster.

**Weaknesses**
- Two minor technical downsides. First is that the sample complexity is a bit off as it incurs an additional O(log^2 n) term. Second is that the time complexity has a dependence on $\sigma$ which does not seem ideal.
- I do find one piece of notation a bit confusing / annoying. When they define $W(H_j) = \max_{H_k \in A} w_k(H_j)$, I think it would be clearer if they had $A$ in the notation of $W$ somehow. Because, as I understand, the set $A$ is built over time but the notation does not contain this. In addition (very minor), maybe make $A$ a set of indices instead so that one can write $\max_{k \in A} w_k(H_j)$.

---

> ### Author Rebuttal · Authors · 2025-07-30
>
> Thank you for your thoughtful review and feedback.
>
> We address your comments below.
>
> ---
>
> ### **Polylogarithmic Sample Complexity (Weakness 1)**
>
> > Two minor technical downsides. First is that the sample complexity is a bit off as it incurs an additional $O(\log^2 n)$ term.
>
> We agree that the gap between our $\operatorname{polylog}(n)$ sample complexity and the non-private $\log(n)$ is an important point.
>
> Our primary contribution is introducing a new stable, private minimum distance estimator that avoids direct comparisons (as in prior tournament-style algorithms) while obtaining a near-linear time private algorithm. To obtain the desired time complexity, our algorithm must *not* make very many wasteful queries, and it must adaptively choose the next hypothesis to query depending on previous outcomes.
>
> Unfortunately, this adaptivity is challenging to privatize because, in principle, the decision of which hypothesis to choose makes the algorithm very sensitive. Thus, we have to make every step private. The additional $\log^2(n)$ factors are a natural byproduct of reconciling the inherent adaptivity of the algorithm with the need for privacy.
>
> Whether these $\operatorname{polylog}$ factors are a necessary price to pay to achieve near-linear time complexity is an interesting open direction. We hope our result paves the way for more progress on this topic.
>
>
> > Second is that the time complexity has a dependence on $1/\sigma$ which does not seem ideal
>
> With respect to $\sigma$, our bound matches those of the $\alpha=3$ non-private algorithm in [ABS24]. While improving these dependencies is a valuable direction for future work (even in the non-private case), our result does not exacerbate these known challenges.
>
> ### **Notation (Weakness 2)**
>
> > I do find one piece of notation a bit confusing / annoying. When they define $W(H_j) = \max_{H_k \in A}w_k(H_j)$, I think it would be clearer if they had $A$
>  in the notation of $W$ somehow. Because, as I understand, the set $A$ is built over time but the notation does not contain this. In addition (very minor), maybe make $A$ a set of indices instead so that one can write $\max_{k \in A}w_k(H_j)$.
>
> Thank you for suggesting a better notation for semi-distances. The notations in our papers were intended to follow the same patterns in previous works like [ABS24] to maintain consistency. However, we will adopt a more clear notation in the final draft.
>
> ---
>
> ### **References**
>
> [ABS24] Maryam Aliakbarpour, Mark Bun, and Adam Smith. Optimal hypothesis selection in (almost) linear time. NeurIPS, 2024.

---

> > ### Comment · Reviewer_MJur · 2025-08-01
> >
> > Thanks for your response. I agree that Weakness 1 is a good direction for future work.
> >
> > I have no further questions.

---

### Note · Authors · 2025-08-12

We sincerely thank all reviewers for their time and effort. We appreciate their positive feedback, as well as their detailed and constructive suggestions for improving this work.

---

### Decision · Program_Chairs · 2025-09-17

**Decision:**

Accept (poster)

**Comment:**

This paper studies a basic task in statistics, known as hypothesis selection, in the differentially private setting. This task has been extensively studied in the standard (non-private setting), where it is almost fully understood---both in terms of sample complexity and computational complexity. The main result of this work is an algorithm that solves this problem in the presence of privacy, with qualitatively near-optimal guarantees. The reviewers agreed that the contribution merits acceptance.